# ATAT 1.1, an Automated Timing Accordance Tool for comparing ice-sheet model output with geochronological data

Jeremy C. Ely[1], Chris D. Clark[1], David Small[2] and Richard C.A. Hindmarsh[3]

[1]Department of Geography, The University of Sheffield, Sheffield, S10 2TN, UK
[2]Department of Geography, Durham University, Durham, DH1 3LE, UK
[3]British Antarctic Survey, High Cross, Madingley Road, Cambridge, CB3 0ET, UK

*Correspondence to*: Jeremy C. Ely (j.ely@sheffield.ac.uk)

**Abstract.** Earth's extant ice sheets are of great societal importance given their ongoing and potential future contributions to sea-level rise. Numerical models of ice sheets are designed to simulate ice sheet behaviour in response to climate changes, but to be improved require validation against observations. The direct observational record of extant ice sheets is limited to a few recent decades, but there is a large and growing body of geochronological evidence spanning millennia constraining the behaviour of palaeo-ice sheets. Hindcasts can be used to improve model formulations and study interactions between ice sheets, the climate system and landscape. However, ice-sheet modelling results have inherent quantitative errors stemming from parameter uncertainty and their internal dynamics, leading many modellers to perform ensemble simulations, while uncertainty in geochronological evidence necessitates expert interpretation. Quantitative tools are essential to examine which members of an ice-sheet model ensemble best fit the constraints provided by geochronological data. We present an Automated Timing Accordance Tool (ATAT version 1.1) used to quantify differences between model results and geochronological-data on the timing of ice sheet advance and/or retreat. To demonstrate its utility, we perform three simplified ice-sheet modelling experiments of the former British-Irish Ice Sheet. These illustrate how ATAT can be used to quantify model performance, either by using the discrete locations where the data originated together with dating constraints or by comparing model outputs with empirically-derived reconstructions that have used these data along with wider expert knowledge. The ATAT code is made available and can be used by ice-sheet modellers to quantify the goodness of fit of hindcasts. ATAT may also be useful for highlighting data inconsistent with glaciological principles or reconstructions that cannot be replicated by an ice sheet model.

## 1 Introduction

Numerical models have been developed which simulate ice sheets under a given climate forcing (e.g. Greve, 1995; Rutt et al., 2009; Pollard and DeConto, 2009; Winkelmann et al., 2011; Gudmundsson et al., 2012; Cornford et al., 2013; Pattyn, 2017). When driven by future climate scenarios, these models are used to forecast the fate of the Antarctic and Greenland ice sheets (e.g. Seddik et al., 2012; DeConto and Pollard, 2016), providing predictions of their potential contribution to future sea level rise. However, incomplete knowledge of ice physics, boundary conditions (e.g. basal topography) and parameterisations of physical processes (e.g. basal sliding, calving), as well as the difficulty of predicting future climate, lead to model-based uncertainty in these predictions (Applegate et al., 2012; Briggs et al., 2014; Ritz et al., 2015). Observations of ice marginal fluctuations (decades) and the processes of ice calving, flow or melting (subaerial or submarine) that facilitate or drive such variations, provide a powerful means to understand the processes leading to the possibility of deriving new formulations that improve

the realism of modelling. However, the short-time span (decades) of these observations limits their being used to
constrain, initialise or validate modelling experiments (Bamber and Aspinall, 2013). Conversely, palaeo-ice
sheets, especially from the last glaciation (~21,000 years ago), left behind evidence which provides the
opportunity to study ice sheet variations across timescales of centuries to millennia, albeit with increased
uncertainty in exact timing.
Numerous modelling studies have aimed to simulate the growth and decay of palaeo-ice sheets, producing
hindcasts of ice-sheet behaviour (e.g. Boulton and Hagdorn, 2006; Hubbard et al., 2009; Tarasov et al., 2012;
Gasson et al., 2016; Patton et al., 2016). Results from these hindcasts may be compared with empirical data
recording ice sheet activity, so as to discern which parameter combinations produce results that best replicate the
evidence of palaeo-ice sheet activity. Three classes of data are of particular use for constraining palaeo-ice sheets;
(i) geomorphological data, (ii) geophysical data, and (iii) geochronological data. Ideally, all three classes of data
should be used to quantify the goodness of fit of a hindcast.
Geomorphological evidence comprises the landforms created by the action of ice upon the landscape, and can
typically provide data on ice extent, recorded by moraines and other ice marginal landforms and on ice-flow
directions recorded by subglacial landforms such as drumlins. Such landforms can be used to decipher the pattern
of glaciation (e.g. Kleman et al., 2006; Clark et al., 2012; Hughes et al., 2014). Two tools, namely Automated
Proximity and Conformity Analysis (APCA) and Automated Flow Direction Analysis (AFDA), have already been
developed which can compare modelled ice margins (APCA) and flow directions (AFDA) to the
geomorphological evidence base (Napieralski et al., 2007).
Geophysical data, in the form of relative sea level measurements and present day uplift rates, provide information
regarding the mass-loading history of an ice sheet. Palaeo-ice-sheet model output is often evaluated against such
data by use of glacio-isostatic adjustment models (e.g. Tushingham and Peltier, 1992; Simpson et al., 2009;
Tarasov et al., 2012; Auriac et al., 2016).
Geochronological evidence attempts to ascertain the absolute timing of ice advance and retreat using dated
material (e.g. organic remains dated by radiocarbon measurement) found in sedimentary contexts interpreted as
indicating ice presence or absence nearby. It enables reconstruction of the chronology of palaeo-ice sheet growth
and decay (Small et al., 2017) and is the underpinning basis for empirically-based ice sheet margin reconstructions
(e.g. Dyke, 2004; Clark et al, 2012; Hughes et al., 2016). Although widely used in empirical reconstruction of
palaeo-ice sheets, geochronological data has rarely been directly compared with ice sheet model output (although
see Briggs and Tarasov, 2013). Such a comparison could be useful both for constraining ice-sheet model
uncertainty and for identifying problems with the geochronological record. For example, a poor fit between model
output and empirical data on timing could inform on the validity of a numerical model (or its parameterisation),
or it could provide a physical basis for questioning the plausibility of empirically-driven interpretations or specific
lines/data points of evidence given that they are associated with inherent uncertainties. In order maximise the
benefit to all users, any comparisons between palaeo-ice sheet model output and empirical data should ideally
consider the inherent uncertainties of both.
Given the wide availability of compilations of geochronological data (e.g. Dyke, 2004; Hughes et al., 2011;
Hughes et al., 2016), as well as the proliferation of ice sheet models (e.g. Greve, 1995; Rutt et al., 2009; Pollard
and DeConto, 2009; Winkelmann et al., 2011; Gudmundsson et al., 2012; Cornford et al., 2013; Pattyn, 2017), a
convenient, reproducible and consistent procedure for comparison should be of great utility to the palaeo-ice sheet

community. The typical volume of geochronological constraints (several thousands) for a palaeo ice sheet and the number of ensemble runs (several hundreds) from an ice sheet model make a visual matching of data and model output nearly impossible to accomplish, which is likely to explain the rarity of such comparisons. Here, we present an Automated Timing Accordance Tool (ATAT, version 1.1). ATAT a systematic means for comparing ice-sheet model output with geochronological data, which quantifies the degree of fit between the two. To separate model uncertainty from data error, a single run of ATAT focuses on the error in geochronological data. This is achieved by comparing geochronological data and its associated error to predictions of ice cover from single ice sheet model output. However, through multiple comparisons against all members from an ensemble ice-sheet modelling experiment, parameter uncertainty can be considered by assessing the degree of fit to the various input parameter combinations. Therefore, ATAT could be used as a basis for examining whether model-data mismatch is a consequence of inadequacies in either the model or data. The tool is in the form of a Python script and requires the installation of open-source libraries. ATAT is written to handle NETCDF data as an input, a format commonly used in ice sheet modelling and is also accessible from many GIS packages in which geochronological data can be stored and manipulated.

## 2 Background

Geochronological evidence and ice sheet model outputs are often independently used to reconstruct the timing of glaciological events. The two approaches are fundamentally different in nature and consequently produce contrasting data outputs. Thus, before describing our approach to comparing the two sets of data (ATAT), we first briefly consider the nature of both geochronological data and ice-sheet model output to highlight the issues and potential difficulties associated with comparing the two and conceptualise a comparison procedure. More extensive descriptions of the nature, uncertainties and limitations of glacial geochronological (Hughes et al., 2016; Small et al., 2017) and model-based (Rougier, 2007; Tarasov et al., 2012; Briggs and Tarasov, 2013) data are considered elsewhere. Given the complex nature of both, those seeking to compare geochronological data and ice-sheet model output should ideally collaborate with those who understand the limitations and uncertainties involved with both forms of data.

### 2.1 Geochronological data

The timing of palaeo-ice sheet activity has primarily been dated using three techniques: (i) radiocarbon dating; (ii) cosmogenic nuclide exposure dating, and (iii) luminescence dating (Figure 1). The utility of each method for determining the timing of palaeo-ice sheet activity has been extensively reviewed elsewhere (e.g. Fuchs and Owen, 2008; Balco, 2011; Small et al., 2017) and only a brief description is provided here. Radiocarbon dating uses the known rate of the radioactive decay of 14C to determine the time elapsed since the death of organic material (Libby et al., 1949; Arnold and Libby, 1951; Figure 1). For palaeo-glaciological purposes, the dated organic material (e.g. shells, mosses, plant remains) is usually taken from basal sediments overlying and closely associated with a glacial deposit in order to determine a minimum deglaciation age (e.g. Heroy and Anderson, 2007; Lowell et al., 2009); ice is interpreted to have retreated from this site some short time prior to this age. Where organic matter is either reworked within or is located directly beneath a glacial deposit, it can be used to constrain the maximum age of glacial advance (e.g. Brown et al., 2007; Ó Cofaigh and Evans, 2007); advance

happened sometime after this age. Cosmogenic nuclides (e.g. 10Be, 26Al and, 36Cl) are produced by the
interaction of secondary cosmic radiation in minerals, such as quartz, within materials exposed at the Earth's
surface (Figure 1). Samples are generally taken from glacially-transported boulders, morainic boulders and
glacially modified bedrock, all of which have ideally had signals from any previous exposure history removed by
glacial erosion. Cosmogenic nuclide dating is thus used to determine the duration of time a sample has been
exposed at the Earth's surface by determination of the concentration of cosmogenic nuclides within that sample.
Luminescence dating can determine the age of a deposit by measuring the charge accumulated within minerals.
This charge accumulates in light-sensitive traps within the crystal lattice due to ionizing radiation produced by
naturally occurring radioactive elements (e.g. U, Th, K). Luminescence dating determines the time elapsed since
the last exposure of the mineral to sunlight; this exposure acts to reset the signal (Figure 1). As subglacial deposits
are unlikely to have been exposed to light before burial, and therefore contain signals accumulated prior to
deposition, luminescence dating within palaeo-glaciology is typically applied to ice marginal sediments, or those
which overly glacial sediments (e.g. Duller, 2006; Smedley et al., 2016; Bateman et al., 2018). All
geochronological techniques record the absence of grounded ice. They therefore provide either maximum or
minimum ages of a glaciological event, depending upon the stratigraphic setting. Table 1 outlines a commonly
used system used to classify geochronological data by stratigraphic setting (Hughes et al., 2011; 2016).
The retreat/advance (ice-free) ages provided by the three geochronometric techniques are all affected by
systematic and geological uncertainties (Small et al., 2017). Systematic uncertainties originate from the tools and
techniques used to derive the date, such as laboratory instruments and sample preparation, and are accounted for
in the quoted errors that accompany a date. Geological uncertainties are caused by the geological history of a
sample, before, during and after a glacial event (e.g. Lowe and Walker, 2000; Lukas et al., 2007; Heyman et al.,
2011). Such influences may leave little or no evidence of their effect upon a sample and are thus hard to quantify.
The relationship between a dated sample and the glacial event it indicates is the largest potential source of
uncertainty in geochronological data and is primarily bounded by the ability of the investigator to find and
associate dateable material to the glacial event of interest. Since all geochronological techniques measure the
absence of ice, expert inference must be made, and are influenced by the availability of information (stratigraphic
or otherwise) at a study site; they may be open to change (e.g. new radiocarbon calibrations, new cosmogenic
isotope production rates). Furthermore, in the cases of luminescence and radiocarbon dating, there can be an
unknown duration since glacial occupation of an area and the deposition of dateable material. These factors mean
it is necessary to consider the quality of dates for ascertaining the timing of the glacial event in question (Small et
al., 2017).
Numerous geochronological studies have sought to ascertain the timing of palaeo-ice sheet activity at sites, leading
to compilations of geochronological data which bring together hundreds to thousands of published dates (e.g.
Dyke et al., 2002; Livingstone et al., 2012; Hughes et al., 2011; 2016). Despite the growing number of reported
dates, they are still insufficient in number and spatial spread to define, on their own, the time-space envelope of
the shrinking ice sheet. Techniques to interpolate geochronological information between sites are required. The
most commonly used technique is empirical ice sheet reconstruction (e.g. Dyke, 2004; Clark et al., 2012), whereby
expert assessments of the geochronological and geomorphological record are used together to create ice-sheet
wide isochrones of ice-sheet margin position and flow configuration. A recent advance in this method has been
the inclusion of confidence envelopes for each isochrone, documenting possible maximum, likely and minimum
extents (Hughes et al., 2016). Further techniques for spatiotemporally interpolating geochronological data include
Bayesian sequence modelling (e.g. Chiverrell et al., 2013; Smedley et al., 2017), in which collections of deglacial
ages are arranged in spatial order determined by a prioi knowledge of geomorphologically-informed ice flow and
retreat patterns (e.g. Gowan, 2013). Such techniques provide viable methods for producing ice-sheet wide
chronologies, filling in information in locations where geochronological data may be sparse.
**2.2 Ice sheet model output**
Ice-sheet models solve equations for ice flow over a computational domain, for a given set of input parameters
and boundary conditions, to determine the likely flow geometry and extent of an ice sheet. Typically, ice-sheet
models run using finite difference techniques on regular grids (e.g. Rutt et al., 2009; Winkelmann et al., 2011).
Ice-sheet models that utilise adaptive meshes (e.g. Cornford et al., 2013) and unstructured meshes also exist (e.g.
Larour et al., 2012) and the results from such models can be interpolated onto spatially regular grids. The spatial
resolution of an ice-sheet model depends upon the computational resources available, and the spatial resolution
of available boundary conditions. Continental-scale models of palaeo-ice sheets have typical spatial resolution of
tens of kilometres (e.g. Briggs and Tarasov, 2013; DeConto and Pollard, 2016; Patton et al., 2016), though parallel,
high-performance computing means higher resolutions are possible (e.g. 5 km in Golledge et al., 2013 and
Seguinot et al., 2016). The temporal resolution of ice sheet model output is ultimately limited by the time-steps
imposed by the stability properties of the numerical schemes solving the ice-flow equations. Given that these
stable time-steps can be sub-annual, output frequency is mostly predetermined by the user (typically decades to
centuries), and as such is constrained by available disk-storage. Ice-sheet models therefore produce spatially
connected predictions of ice-sheet behaviour such as advance and deglaciation (e.g. Table 1) across gridded
domains at various temporal and spatial resolutions.
The stress fields imposed upon ice can be fully described by solving the Stokes equations. Indeed, 'full Stokes'
models which do so have been tested (Pattyn et al., 2008) and used to simulate ice sheets (e.g. Seddik et al., 2012).
However, fully solving the Stokes equations over the spatio-temporal scales relevant to palaeo-ice sheet
researchers remains beyond the limit of currently available computational power. This problem is exacerbated by
the need to run multi-parameter valued ensemble simulations to account for model uncertainty over multi-
millennial and continental-scale domains.  This means that palaeo-ice sheet modelling experiments rely upon
approximations of the Stokes equations (see Kirchner et al., 2011 for a discussion), such as the shallow ice
approximation (SIA) and shallow shelf approximation (SSA). The choice of ice-flow approximation used within
a model has implications for the capability of models to realistically capture aspects of ice sheet flow (Hindmarsh,
2009; Kirchner et al., 2011; 2016), and in turn influences the nature of the model output produced. For instance,
the SIA is not applicable for ice shelves, therefore SIA-based models do not produce modelled ice shelves (e.g.
Glimmer; Rutt et al., 2009). Therefore, the timing of deglaciation in an SIA model can be determined as the point
at which ice thickness in a cell becomes zero or thinner than the flotation thickness, whereas in a SSA or higher-
order model the location and movement of the grounding line must be determined.
Though ice sheet models produce output which is consistent with model physics, like all numerical models of
physical systems (e.g. Rougier, 2007) there are many sources of uncertainty involved with ice sheet modelling.
Three broad sources of model-based uncertainty can be distinguished: (i) down-scaling; (ii) parametric
uncertainty; (iii) structural uncertainty. These are defined and discussed below.
Down-scaling uncertainties arise due to an ice-sheet models computation over space which has a coarser resolution
than reality. This means that a characteristic which can be measured to a high level of accuracy and precision for
a real ice-sheet (e.g. the position of a calving front), has a larger uncertainty in an ice-sheet model. This is
especially pertinent for data-model comparisons, as most observations of ice-sheet activity have a sub-model
resolution.
Parametric uncertainty has two main sources: (i) parameterisations, and (ii) boundary conditions. Where a process
is too complex (e.g. calving) or occurs at too small a scale (e.g. regelation) to be captured by an ice sheet model,
it is often simplified and parameterised. Associated with each parameterisation are a set of parameters, the values
of which are either unknown, or thought to vary within some plausible bounds, and which can either be constant
or spatially and temporally variable across a domain. An example of a process which is often parameterised is
basal sliding. This parameterisation is often done through the implementation of a sliding law (e.g. Fowler, 1986;
Bueler and Brown, 2009; Schoof, 2010), which relates the basal shear stress to the basal velocity (Fowler, 1986).
Parameters used to determine this relationship are often assigned or incorporated within a parameter, or prescribed
by another model parameterisation (e.g. a subglacial hydrology model). Adding to the uncertainty in the absence
of a single preferable sliding law, ice-sheet models often allow the user to choose between different sliding law
implementations.
Boundary conditions, the values prescribed at the edge of the modelled domain, also introduce uncertainty into
ice-sheet models. For contemporary ice sheets, there is a large uncertainty in the basal topography (e.g. Fretwell
et al., 2013). This is less of a problem for the more accessible beds of palaeo-ice sheets. However, accurately
accounting for the evolution of this bed topography over the course of a glaciation requires a model of isostatic
adjustment (Lingle and Clark, 1985; Gomez et al., 2013).
A very large source of uncertainty for modelling palaeo-ice sheets is the climate used to drive them (Stokes et al.,
2015), as indeed is the case for forecasts of contemporary ice sheets (e.g. Edwards et al., 2014). Owing to the
computational resources required and technical challenges, few palaeo-ice sheet models are coupled with climate
models. This uncertainty over past climate is reflected in the large range of outputs produced by global circulation
models which have tried to simulate the last glacial cycle (e.g. Braconnot et al., 2012). Palaeo-ice sheet modellers
have used a range of methods to force their models, including simple parameterisations (Boulton and Hagdorn,
2006), applying offsets derived from ice core records to contemporary climate (e.g. Huybrechts, 1990; Hubbard
et al., 2009) and scaling between present-day conditions and uncoupled global-circulation-model simulations at
maximum glacial conditions (e.g. Greve et al., 1999; Gregoire et al., 2012; Gasson et al., 2016). Each approach is
associated with an inherent uncertainty. When this uncertainty is accounted for in an ensemble experiment, the
range of possible climates produces numerous ice sheet outputs.
Structural uncertainty is related to parametric uncertainty, but has a broader remit, and is defined as uncertainty
which occurs due to differences in model coding and design (Collins, 2007; Tebaldi and Knutti, 2007). This
encompasses differences in which processes are included in different models, and also the manner in which they
are implemented. Structural uncertainty is difficult to quantify, but can be explored by multi-model comparison
(Murphy et al., 2004; Collins et al., 2011). Such comparisons are not currently routine in palaeo-ice sheet
modelling. Differences in model coding (i.e. structural uncertainty), arise due to a lack of understanding regarding
the physical system in question. This points to a broader uncertainty with a similar remit, that no models can
include processes that are as yet unknown to science. Reducing this source of uncertainty is an ongoing challenge
for glaciology.

There is another uncertainty  which hinders ice-sheet models from being able to accurately predict the evolution
of ice-sheets, which is the presence of instabilities – we use this term in the technical sense of a small perturbation
that leads to the whole ice-sheet system amplifying this small perturbation to the extent it can leave a mark in the
geological record. A classic example of this in ice-sheet dynamics is the marine ice-sheet instability (MISI), first
discussed in the1970s (Hughes, 1973; Weertman, 1974, Mercer, 1978) and more recently put on a sounder
mathematical footing (Schoof 2007, 2012).
The MISI actually refers to an instability in grounding-line (GL) position on a reverse slope, where the water
depth is shallowing in the direction of ice flow. Since ice flux increases with ice thickness, a straightforward
argument leads to the conclusion that if the GL advances into shallower water, the efflux will decrease, the ice
sheet will gain mass and the advance continue. If, on the other hand, the GL retreats, the flux will increase, the
ice-sheet will lose mass and the retreat continue. In principle, given the right parameterisations and basal
topography, ice-sheet models should be able to predict the 'trajectory' of GL migration arising as a consequence
of the MISI. However, the MISI is one of the class of instabilities that lead to poor predictability; certain small
variations of parameters and specifications will lead to large-scale changes in the 'trajectory', in this case the
retreat history. A well-known analogy is the 'butterfly effect', which originated in atmospheric modelling work
(Lorenz, 1963); the butterfly effect is concerned with the consequences of the statement "small causes can have
larger effects". Recent work has also shown that additional physical processes, such as ice-shelf buttressing
(Gudmunsson, 2012) and the effect that the gravitational pull of ice-sheets has on sea level (Gomez et al., 2012)
have additional effects on grounding line stability. Given that most of the palaeo-ice sheets during the last glacial
cycle had extensive marine margins and overdeepened basins, with isostatic adjustment creating further zones of
reverse slope, capturing grounding line processes is important for simulating these ice-sheets.

**2.3 Considerations when comparing geochronological data and ice-sheet model output**

Sections 2.1 and 2.2 make it clear that several factors must be considered in order to satisfactorily compare
geochronological data and ice-sheet model output (Table 2). Most critically, the two datasets involved in any
comparison have varying spatial properties. Raw geochronological data is unevenly distributed and located at
specific points, with horizontal position accurate to a metre or so; such data may be used to plot ice-margin
fluctuations of the order of tens of kilometres (Figure 2C). Ice-sheet models typically produce results on evenly-
spaced points (at ~5 km to 20 km resolution) that are distributed over and beyond the maximum area of the palaeo-
ice sheet (Table 2; Figure 2B). Consequently, in comparing the two, a choice must be made; either
geochronological data should be gridded (coarsened) to the resolution of the ice-sheet model, or the ice-sheet
model results must be interpolated to a higher resolution. Both options have drawbacks, as the former removes
spatial accuracy from geochronological data while the latter relies upon interpolation beyond model resolution
and, more seriously, model physics. A second problem lies in the spatial organisation of the data (Table 2). Ice-
sheet models produce a regular grid of data (Figure 2B), meaning that no location is more significant than any
other when comparing the modelled deglacial chronology with that inferred from geological data. Conversely,
owing to the uneven distribution of raw geochronological data, some regions of a palaeo-ice sheet may be better
constrained than others (Figure 2C). As noted by Briggs and Tarasov (2013), any comparison that does not treat
the uneven spatial distribution of geochronological data may favour sites where numerous dates exist over more
isolated locations. One approach to overcoming these disparities is to use an interpolation scheme (e.g. empirical
reconstruction, Bayesian sequence) on the raw geochronological data. This produces a geochronological
framework by combining evidence on pattern and timing to yield a distribution that is spatially more uniform and
a spatial resolution similar to that of palaeo-ice sheet model output (Figure 2D).
The temporal intervals between and precision of geochronological data and ice sheet model output also vary
(Table 2). The time intervals between geochronometric data are determined by the number of available
observations, and precision determined by sources of uncertainty. Conversely, ice sheet models produce output at
regular intervals and are temporally exact, which is to be contrasted with 'correct'. Since the output interval of an
ice-sheet model is generally determined by the user (see Section 2.2) it is pertinent to consider an appropriate
time-interval of ice-sheet model output for comparison with geochronological data. For example, radiocarbon
dates have precision typically in the order of hundreds of years but do not directly constrain ice extent, whilst
empirically reconstructed isochrones are typically produced for thousand-year time-slices (e.g. Hughes et al.,
2016). In reality, ice-sheets may respond to events at faster time-scales than this, but in the absence of internal
instabilities (e.g. MISI) palaeo-ice sheet models are ultimately limited by the temporal resolution of the available
climate forcing data. Thus, to gain insight into controls on palaeo-ice sheet behaviour, it may be necessary to
create model output with a greater (centurial) temporal resolution than the uncertainty associated with
geochronology.
Both geochronological data and ice-sheet model output have sources of uncertainty which must also be considered
when comparing the two. For geochronological data, uncertainty is typically expressed as a standard deviation
from the reported age, and are therefore easy to consider when comparing to an ice sheet model. For ice-sheet
models, individual model runs do not currently express uncertainty, and it is only when multiple, ensemble, runs
which systematically vary parameters and boundary conditions are conducted that uncertainty in all output
variables can be expressed. Therefore, any comparison between geochronological data and model simulations
must either compare to all members of an ensemble experiment in turn, or against amalgamated output from an
ensemble which considers model uncertainty. Having said this, statistical techniques exist to derive probability
distribution functions for individual quantities (e.g. Ritz et al., 2015). Such ensemble runs typical comprise
hundreds to thousands of individual runs (Tarasov and Peltier, 2004; Robinson et al., 2011). Given the volume of
data this produces, one appealing application of a quantitative comparison between geochronological data and ice
sheet model output would be to act as a filter for scoring ice-sheet model runs and reducing predictive uncertainty
by only using the parameter combinations that were successful. However, if all possible parameters have been
modelled, (i.e. the full 'phase-space' of the model has been explored (cf. Briggs and Tarasov, 2013)), and very
few (or no) model runs conform to a certain set of geochronological data or an empirical reconstruction, this may
provide a basis to question aspects of the evidence (e.g. re-examining the stratigraphic context of a dated sample
site or questioning the basis of the reconstructed isochrone). Of course, a third possibility that both data and model
are incorrect cannot be excluded.
We therefore suggest that any comparison between ice-sheet model experiments and geochronological data should
consider:
i) That both ice-sheet models and geochronological data have inherent uncertainties;

ii) That geochronological data typically provide a constraint on just the absence of ice; such that ice must have withdrawn from a site sometime (50 years? 500 years? 5000 years?) prior to the date (which can be any point within the full range of the stated uncertainty). It is thus a limit in time and not a direct measure of glacial activity. Figure 3 illustrates this for advance and retreat constraints. It is most often the case that dated material is taken close to the stratigraphic boundary or landform representing ice presence, in which case a date might be considered as a 'tight constraint' (e.g. the ice withdrew and very soon afterwards (50 years) marine fauna colonised the area and deposited the shells used in dating). Sometimes however there may have been a large (centuries to millennia) interval of time between the withdrawal and the age of the shell chosen as a sample, in which case the date will provide a 'loose' limiting constraint; it might be much younger than ice retreat (Figure 3).

iii) There is inherent value to the expert interpretation of stratigraphic and geomorphological information, meaning an ice-free age reported for a site is likely as close as possible (tight constraint) to a glacial event. However, this interpretation could be subject to change;

iv) Geochronological data exist as spatially distributed dated sites (e.g. Figure 2C) which can be built into a spatially coherent reconstruction (e.g. Figure 2D);

v) A great input uncertainty in a palaeo-ice sheet model is the climate, which can lead to changes in the spatial extent and timing of ice sheet activity.

vi) A factor which requires further investigation is the relationship between the operation of a physical instability (e.g. the MISI) and the practical ability of models to predict retreat or advance rates; the presence of an instability can result in extreme sensitivity to parameter ignorance or over-simplified model physics.

vii) Other uncertainties can also lead to variations in ice-sheet model results; these can be accounted for in an ensemble of hundreds to thousands of simulations.

Given the above, it is unlikely that a single procedure could capture model-data conformity. ATAT therefore implements several ways of measuring data-model discrepancies and produces output maps (described in the following two sections) to help a user assess which model runs best agree with the available geochronological data. One approach is to transform the geochronological data points (x,y,t) to a gridded field (raster) that define age constraints of ice advance and another grid for retreat . Both of these data types also require an associated grid that reports the uncertainty range as error (Figure 4). These age grids may then be quantitatively compared to equivalent grids (age of advance grid and age of retreat grid) derived from the ice sheet model outputs. Alternatively, one might prefer to compare model runs against the geochronological data (points) combined with expert-sourced interpretive geomorphological and geological data, in which age constraints from dated sites have been spatially extrapolated using moraines and the wider retreat pattern. In this case ATAT allows the model outputs to be compared to the 'lines on maps' type of reconstruction subsequent to conversion from age isolines to a grid of ages (Figure 4).

**3. Description of tool**

ATAT is written in Python, and utilises several freely available modules. Access to these modules may require a Python package manager, such as 'pip' or 'anaconda'. ATAT can therefore be run from the command line on any operating system, or by using a Python interface such as IDLE.

**3.1 Required data and processing**

ATAT requires two datasets as an input: (i) an ice-sheet model output; and (ii) gridded geochronological data. Table 3 provides the required variables and standard names for each dataset. In order to determine the advance age or deglacial age predicted by the ice sheet model, ATAT requires either an ice thickness (where the model does not produce ice shelves) or a grounded ice-mask variable (where ice shelves are modelled). In the latter case, the user is asked to define the value which represents grounded ice.

Empirical advance and deglacial geochronological data (Table 1) require separate input files (NETCDF format), as model-data comparison for these two scenarios are run separately in ATAT. Table 1 and further references (Hughes et al., 2011; 2016; Small et al., 2017), provide information regarding identification of the stratigraphic setting of these two glaciological events as considered by ATAT. ATAT requires that geochronological data (advance or deglacial) are interpolated onto the same grid projection and resolution as the ice-sheet model before use. Though an imperfect solution to the problem of comparing grids of different resolution, (Section 2.3; Table 2), this was preferred to the alternative solution of regridding an ice sheet model onto a higher resolution grid, as this may introduce the false impression of high resolution modelling sensitive to boundary conditions (e.g. topography) beyond the actual model resolution.

Preparation of the geochronological data to be the same format and grid resolution as the ice sheet model output requires use of a GIS software package such as ESRI ArcMap or QGIS. Users must define deglacial/advance ages based either upon the availability of geochronological data in a cell, or based upon an empirical reconstruction (Figure 4). These ages must be calibrated to a calendar which is the same as that output by the ice-sheet model (in our case the 365-day calendar in units of seconds since 1-1-1). Where there are no data (i.e. outside the ice-sheet limit), the grid value must be kept at 0. When multiple dates are contained within a cell, expert judgement is required to ascertain which date is most representative of the deglaciation of a region. This assessment should be based upon the quality of sample taken; criteria for establishing this quality are considered in Small et al. (2017). In the case where a profile of dates has been collected (for example up a vertical section at the side of a valley, or from multiple depths of a marine core) the date which most closely defines the timing of final deglaciation of an area should be chosen, as this is the focus of ATAT. The assembly of this geochronological database input into ATAT should consider the reliability of ages, removing outliers and unreliable ages (see Small et al. (2017) for a discussion of this issue). In particular, loose constraints, such as cosmogenic dates which display inheritance or radiocarbon dates effected by a depositional hiatus, should be removed as this have the potential to bias results. In a comparable manner, the attribution of error to each cell is also reliant upon expert interpretation. The magnitude of error may vary between the source of geochronological data (radiocarbon, cosmogenic nuclide or luminescence) and user choice for experimental design (e.g. 1, 2 or 3 sigma). A single error value must be given for each dated cell, corresponding to the maximum threshold beyond which the user deems it is unacceptable for a model prediction to occur (Figure 3). Given that creating this input data may involve many expert decisions (e.g. which date has the relevant stratigraphic setting, which date(s) are most reliable?), this part of the process is not yet automated within ATAT. This data preparation stage is therefore the most time-consuming and user-intensive part of the process. However, users only need to define the data-based advance/deglacial grid once to compare to multiple model outputs. Future work should consider alternatives means of choosing dates and identifying outliers, such as Bayesian age modelling (e.g. Chivverell et al., 2013). The input data NetCDF file should also contain the

variables latitude, longitude, base topography (the topography that the ice-sheet modelling is conducted on and the elevation of the geochronological sample (Table 3).

ATAT is called from a suitable python command-line environment, using several system arguments to define input variables (Table 1; Figure 5). Users must define whether they are testing a deglacial or advance scenario. ATAT only considers the last time that ice advances over an area. Therefore, caution must be undertaken when defining advance data in regions where multiple readvances occur, and users should consider limiting the time interval of the ice sheet model tested when examining specific events (e.g. a well-dated readvance or ice sheet build-up). The location of the file containing the geochronological data grid (e.g. Figure 5) is then required. From this file, the age and error grids are converted to arrays. For the age data, null values are masked out using the numpys masked array function. A second array that accounts for error is then created, the properties of which depends upon whether a deglacial or advance scenario is being tested. For a deglacial scenario, a model prediction will be unacceptable if the cell is ice-covered after the range of the date error is accounted for, but the cell may become deglaciated any time before this. Therefore, the associated error value is added onto the cell date, to create a maximum age at which a cell must be deglaciated by to conform to the ice sheet model (Figure 3). The opposite is true for advance ages; ice can cover a cell any time after the date and associated error, but cannot cover the cell before the date of the advance. In order to allow for advances which occur after the date and its error, associated error is therefore subtracted from the date cell (Figure 3). To account for the uneven spatial distribution of dates, a weighting for each date is then calculated based upon their spatial proximity. This weighting is used later when comparing the data to the model output. To calculate this weighting ($w_i$), ATAT defines a local spatial density of dated values based upon a kernel search of 10 neighbouring cells.

The user must define the path to the ice sheet model output, from which the modelled deglacial age will be calculated and eventually compared to the data (Figure 4). The user must also define whether to base deglacial timing on an ice thickness or grounded extent mask variable (Table 2). If the user selects thickness, the margin is defined by an increase from 0 ice thickness. For the mask, the user is also asked to supply the number which refers to grounded ice extent. The timing of advance is then determined by the change of a cell to this number (Figure 5). The margin position recreated by the ice-sheet model has a spatial uncertainty due to downscaling issues and fluctuations which may occur between recorded outputs. To account for this, ATAT calculates a second set of modelled deglacial ages, whereby the deglaciated region at each modelled time output is expanded to all cells which neighbour the originally identified deglaciated or advanced over cells. Furthermore, the spatial resolution of ice-sheet models typically means that the emergence of ice-free topography at the edge or within an ice-sheet (e.g. in situations such as steep-sided valleys or nuntaks) are poorly represented. To account for this, ATAT firstly calculates the modelled ice-sheet surface at each time output by adding ice thickness to the input base topography. Where the modelled surface elevation is below that of the sample elevation, these cells are identified as being deglaciated (Figure 5). The downscaling of topography onto ice-sheet model grids also introduces a vertical uncertainty. This is accounted for in ATAT through calculating the difference between sample elevation and the reference elevation. A second metric which identifies cells as having been deglaciated if they are also within this vertical uncertainty is also calculated (Figure 5).

**3.2 Model-data comparison**

Once the required variables have been retrieved from the NETCDF data and manipulated, ATAT compares the geochronological age and modelled age at each location (Figure 4). Firstly, the grid cells which have data are categorised as to whether there is model-data agreement, based on the criteria shown in Figure 3. Since all dating techniques only record the absence of ice, geochronological data provides only a one-way constraint on palaeo-ice sheet activity. For deglacial ages, deglaciation could occur any time before the geochronological data provided and within the error of the date (i.e. deglacial ages are minimum constraints), but deglaciation must not occur after the error of the date is considered (Figure 3). For advance ages, advance must have happened after the date or within error beforehand (i.e. advance ages are maximum constraints), but palaeo-ice sheet advance cannot occur in the time period before that dated error (Figure 3). Once ATAT has determined whether each cell conforms to these criteria, a map is produced identifying at which locations the ice sheet model agrees with the geochronological data.

Though the criteria described above and illustrated in Figure 3 allow for the identification of dates which conform to the predictions of an ice sheet model, they provide little insight into how close the timing of the model prediction is to the geochronological data. If these were the only criteria on which a model-data comparison was made, it could prove problematic. In an extreme case, one could envisage that all retreat dates are adhered to by a model run that deglaciates from a maximum extent implausibly rapidly (say 50 years!), and, given that we only have one-way (minimum) constraints on deglaciation (Figure 3), this model run would conform to all modelled dates. Whilst the nature of geochronological data (being only able to determine the absence of ice) does not preclude such a scenario, this assumes that there is no inherent value to the expert judgement and stratigraphic interpretation of each date as being close to palaeo-ice sheet timing (cf. Small et al. 2017). Therefore, ATAT also determines the temporal proximity of the geochronological data and the model prediction. Firstly, a map of the difference between modelled and empirical ages is created (Figure 5). This enables the identification of dates which are a large distance away from the model prediction. Secondly, the route-mean square error (RMSE) is calculated using the Eq. (2):

$$RMSE = \sqrt{\frac{1}{n}\sum_{i=1}^{n}(g_i - m_i)^2} \ ,$$

(1)

where n is the number of cells which contain empirical geochronological information, $g_i$ is the associated geochronological date, and $m_i$ is the model predicted age. The RMSE works well when the geochronological data is evenly spatially distributed, either from a reconstruction (i.e. isochrones) or a wealth of dates. ATAT also calculates a weighted RMSE (wRMSE), for situations where this is not the case (i.e. there is a paucity of dates that are not distributed evenly across the domain) using Eq. (3):

$$wRMSE = \sqrt{\frac{1}{n}\sum_{i=1}^{n}((g_i - m_i)/w_i)^2} \ ,$$

(2)

where $w_i$ is the spatial weighting factor. Results of the RMSE and wRMSE calculations are separated by the degree to which included dates agree with model output. This creates an array of metrics with varying levels of consideration of model and data uncertainty (Figure 5). Both the RMSE and wRMSE are calculated for all dates,

to create a metric that doesn't account for dating error but may give an indication of how close a model-run gets
to dated cells. Dated locations are also categorised according to whether model-data agreement occurs within
dating error, and whether the addition of horizontal (ice margin) and vertical (ice surface) downscaling uncertainty
means that model-data agreement occurs. The RMSE and wRMSE are calculated for these categories to create a
metric which accounts for data and model uncertainty (Figure 5). ATAT then produces a .csv file containing all
calculated statistics per ice-sheet model output file. We suggest that the most rigorous metric, the wRMSE of
dates which conform within geochronological data and model downscaling uncertainty (Figure 5), should most
frequently used. However, other metrics, such as the RMSE of all dates, may give an indication of performance
earlier in the modelling process. For example, initial results may reveal that no or very few dates conform to a set
of model simulations within model and data uncertainty, but the RMSE of all dates may give an indication of
models and associated parameters to be explored further. Given the complexity of data-model comparison,
different statistics may have different uses. For instance, the percentage of covered dates may prove useful to
identify the worst performing model runs (i.e. the bottom 50%), whilst the wRMSE of dates within error may be
more convenient for choosing between model runs. However, given the uncertainty in ice-sheet modelling it is
likely that in an ensemble there will be no single model run which has significantly better metrics than others, so
ATAT may best be used to choose members which pass a user-defined threshold of combined metrics.
Pragmatically, we envisage that ATAT could be used in the following ways, though others may exist. In sensitivity
experiments (e.g. Huybrechts, 1990; Hubbard et al., 2009; Patton et al., 2016), ATAT could be used to quantify
how the alteration of a parameter influences the fit of a model to geochronological data. In ensemble experiments,
ATAT could be used to rank the performance of individual ensemble member simulations with respect to
geochronological constraints, either as a means of ruling out simulations with the poorest performance (e.g.
Gregoire et al., 2012) or calibrating input parameters for further experiments (e.g. Tarasov et al., 2012). Where
the results of an ensemble experiment have been amalgamated (i.e. where each cell has a distribution of ice-free
ages), ATAT could be compared to measures of average modelled deglaciation/advance age and against standard
deviations of these. Such comparisons could reveal areas of persistent model-data mismatch. If this is the case,
this may form the basis of identifying regions of significant model uncertainty (does this site not match due to
poor implementation of processes in the model?) or form the basis for re-examination of the geological evidence
(are there reasons why this site is consistently an outlier?). Furthermore, ATAT could be used to explore how
incorporating additional processes into a model alter the fit to data. Here, we envisage two sets of model
experiments, one which includes a new implementation of a process in a model and another which does not
implement this process, whilst holding all other things equal between the two experiments. ATAT could then be
used to distinguish whether a better fit to geochronological data can be made when the new process is accounted
for.
**4. Application of tool**
**4.1 Ice Sheet Model**
To trial ATAT we used geochronological data and ice sheet modelling experiments from the former British-Irish
Ice Sheet (BIIS). A vast quantity of previous research has produced a high density of dates (Hughes et al., 2011)
which are being substantially augmented by the BRITICE-CHRONO project (http://www.britice-
chrono.group.shef.ac.uk/). Along with an abundance of well documented landforms (Clark et al., 2017), this
makes the BIIS a data-rich study area for empirical reconstructions and ice sheet modelling. Ongoing modelling
work aims to capture the behaviour of the BIIS inferred from the geomorphological and geochronological record
(see Clark et al., 2012 for a recent reconstruction). We do not expect our model to capture these specific details.
Instead, the purpose of modelling in this paper is merely to illustrate the use of ATAT. We therefore restrict
ourselves to simplified modelling experiments and show only three model runs (Experiments A, B and C), whereas
a full ensemble experiment would contain hundreds or thousands of simulations.
Ice sheet modelling experiments were conducted using the Parallel Ice Sheet Model (PISM; Winkelmann et al.,
2011). This is a hybrid SIA-SSA model, with an implementation of grounding line physics. It is therefore suited
to modelling both the marine-based portions of the BIIS and the terrestrial realm. The model simulates the history
of the BIIS from 40 ka to present. The model is run at 5 km resolution, with basal topography derived from the
General Bathymetric chart of the Oceans (www.gebco.net). This is updated to account for isostatic adjustment
using a viscoelastic Earth model (Bueler et al., 2007) and a scalar eustatic sea level offset based on the SPECMAP
data (Imbrie et al., 1984). All three model runs, labelled A-C, had the same input parameters and boundary
conditions, apart from climate forcing. We take a similar approach to Seguinot et al. (2016) in computing a climate
forcing. Modern values of temperature and precipitation are perturbed by a proxy temperature record, in this case
the GRIP ice core record (Johnsen et al., 1995). These are input into a positive degree day model to calculate mass
balance (Calov and Greve, 2005). Input precipitation values are the same between experiments. To introduce
variation between the experiments, temperature varies such that Experiment A is the equivalent of modern day
values, Experiment B has values uniformly reduced by 1°C and Experiment C has values uniformly reduced by
2°C. All other parameters and forcings are equal between experiments. This simple approach to climate forcing
here used for demonstration purposes only, and does not capture the changes to atmospheric and oceanic
circulation patterns that occur during a glacial cycle.
The maximum extent of ice for each experiment is shown in Figure 6 and the timing of advance and retreat is
shown in Figure 7. Potentially unrealistic ice sheets occur in the North Sea, perhaps due to the choice of domain
not including the influence of the Fennoscandian ice sheet in this area. As noted above, we do not expect these
model runs to fully replicate the reconstructed characteristics of the BIIS (e.g. Clark et al., 2012). However, it is
worth noting general, visually-derived, observations regarding the outputs shown in Figure 6. For larger
temperature offsets, the ice sheet gets bigger, the timing of maximum extent gets progressively later and the
modelled ice sheet gets thicker (Figure 6). In all experiments, there is generally a gradual advance toward the
maximum extent followed by retreat (Figure 7). This pattern is interrupted by a later readvance that corresponds
to the timing of the Younger Dryas in the GRIP record; this causes ice to regrow over high elevation areas such
as Scotland and central Wales. The extent of this readvance increases with decreased temperature offsets between
experiments (Figure 7). Smaller readvances, occurring around 16.5 ka also occur (Figure 7).
**4.2 Geochronological data**
Ice-sheet advance dates were taken from the compilation of Hughes et al. (2016) and gridded to the ice sheet
model domain (Figure 4). In total, 61 cells were represented with advance dates (Figure 8A). Considering now
ice-sheet retreat (Figure 8B), dates deemed reliable or probably reliable by Small et al. (2017) were used (i.e.
those given a 'traffic light rating' of green or amber). For the dated advance and retreat locations, the
geochronological data in each cell was assigned an error corresponding to that which was reported in the literature.
We also compared our results to the 'likely' empirical reconstruction of Hughes et al. (2016), based on that of
Clark et al. (2012) (Figure 8C), using the minimum and maximum bounding envelopes to assign an error to each
cell of the ice sheet grid (Figure 8D). The largest errors occur in the North Sea region, where there is a lack of
empirical data (e.g. Figures 8A and B).

## 4.3 Results

Table 4 shows selected statistics derived by ATAT when comparing the three ice-sheet modelling experiments
(Figures 6 and 7) against the three categories of data (Advance, Retreat, Isochrones; Figure 8). wRMSE was not
calculated for the DATED isochrone reconstruction, as grid points are distributed evenly and therefore have equal
spatial weighting (Table 4). Experiment C produces modelled ice-sheets with the greatest areal extent, and
therefore performs best at correctly covering the dated areas (Table 4). However, none of the three experiments
perform particularly well when compared with the data or the empirical reconstruction regarding timing and
results in high (>2000 year) RMSEs (Table 4). The application of ATAT and the results from these simplified
experiments allow us to suggest directions for analysing future experiments.
All three experiments produced large RMSEs, in the order of thousands of years, when compared to all three
categories of data (Table 4). For advance ages, the three simulations conform to a large number of dated locations
(e.g. 72% of ages in Experiments B and C; Table 4). However, the RMSEs of advance ages are high (Table 4).
This shows that, while the models perform well at matching the constraint of covering an area in ice after an
advance age (Figure 3), the models often glaciate a region much later than required. Advance dates are particularly
difficult to obtain from the stratigraphic record, and often there may be a long hiatus between the initial deposition
of datable material and the subsequent advance of a glacier. Future experiments with large ensembles should
therefore consider the number of advance dates conformed to (rather than the RMSE) as a more robust guide for
model performance during ice advance.
For the retreat comparisons, the three modelling experiments conform to a larger percentage of sites, seemingly
outperforming the empirically-derived DATED reconstruction (Table 4). However, where model-data agreement
occurs, the RMSE produced are much higher when the model is compared to the DATED reconstruction. This is
due to the reconstruction containing large uncertainties in regions which lack geochronological control (for
example in the North Sea, Figure 8). These uncertainties, a product of spatial interpolation across regions with
sparse information, are much greater than those associated with individual dates. Figure 9A shows examples of
output maps from ATAT which display the spatial pattern of agreement and the magnitude of the difference
between Experiment C and the DATED reconstruction. This shows that due to the uncertainty associated with
North Sea glaciation, even where the model produces an unrealistic artefact, there is data-model agreement.
Furthermore, ATAT produces a map which displays the number of years between data-based and modelled retreat
and/or advance (e.g. Figure 9B). Figure 9B, which compares Experiment C to the DATED isochrones, shows that
the timing of model-data disagreement is spatially variable. If more modelling simulations were conducted, such
maps may reveal regions of reconstruction or particular dates which are difficult to simulate in the model. In such
cases, data or model re-evaluation may be required and herein lies the potential utility of this ATAT tool in making
sense of ensemble model runs. However, such model-data comparison awaits a full-ensemble simulation which
accounts for model uncertainty (e.g. Hubbard et al., 2009).

**5. Summary and concluding remarks**

Here we present ATAT, an automated timing-accordance tool for comparing ice-sheet model output with geochronological data and empirical ice sheet reconstructions. We demonstrate the utility of ATAT through three simplified simulations of the former British-Irish Ice Sheet. Note that a larger ensemble model of hundreds to thousands of runs is required for model evaluation (e.g. Hubbard et al., 2009). ATAT enables users to quantify the difference between the simulated timing of ice sheet advance and retreat and those from a chosen dataset, and allows production of cumulative ice coverage agreement maps that should help distinguish between less and more promising runs. We envisage that this tool will be especially useful for ice-sheet modellers through justifying model choice from an ensemble, quantifying error and tuning ice-sheet model experiments to fit geochronological data. Ideally, this tool should be used in combination with other evaluation methods, such as fit to relative sea-level records. In the case where locations or regions of data cannot be fit by a model, and all model uncertainty has been accounted for in an ensemble simulation, the comparisons made in ATAT may also highlight that data re-evaluation is necessary. ATAT is supplied as supplementary material to this article.

**6. Code Availability**

ATAT 1.1 source code is freely distributed under a GNU GPL licence as supplementary material to this paper. It can also be downloaded with exmple input grids from https://figshare.com/s/6c8f885e9d10558ed359. An example geochronological data grid and ice-sheet model grid can also be downloaded from this link. The ice sheet modelling experiments shown here were conducted using the Parallel Ice Sheet Model (http://pism-docs.org/). Development of PISM is supported by NASA grant NNX17AG65G and NSF grants PLR-1603799 and PLR-1644277. The geochronological data used is freely available from https://www.sciencedirect.com/science/article/pii/S0012825216304408#s0105 and https://doi.pangaea.de/10.1594/PANGAEA.848117.

**6.1. General Instructions**

ATAT is written in python, and distributed as both .py script, for use in Python 2, and a .py3 script, for use with Python 3. The tool requires instillation of Python and the following freely available Python packages:

- `netCDF4` (https://pypi.python.org/pypi/netCDF4)
- `numpy` (http://www.numpy.org/)
- `scipy` (https://www.scipy.org/)
- `matplotlib` (https://matplotlib.org/)
- `matplotlib toolkit basemap` (https://matplotlib.org/basemap/)

ATAT can be run from any Python enabled environment (e.g. IDLE, BASH). Here we provide the following simple instructions for running ATAT in a BASH shell. For numerous runs, a shell script should be created.

From the command line, launch the ATAT script using python ("python ATATv1.1.py"). Eight command-line arguments (A1 - A8), separated by a space should then follow.

A1 dictates whether deglacial or advance ages are being tested. Type "DEGLACIAL" or "ADVANCE" accordingly.

A2 is the path to the geochronological data file (e.g. "/home/ATAT/geochron.nc")

A3 defines whether the model extent is based on thickness or a mask. Type THK or MSK accordingly.
A4 is the path to the ice-sheet model output file (e.g. "/home/ATAT/icesheetmodel1.nc")
A5 is the value of the ice-sheet output mask. A value is required even if A3 = THK, but can be any value as it will
be ignored.
A6 to A8 control output maps. A6 defines whether the output map should consider margin uncertainty, with a
value of BORDER or NONE.
A7 defines whether the model-data offset map displaces RMSE (option "NONE") or wRMSE ("WEIGHTED").
A8 specifies which dates are plotted on the difference map, and can be "ALL" for all dates, "COVERED" for
those which at some point where covered by ice and "INERROR" to display only those dates where model-data
agreement within dating error occurred.
An example command would be "python ATATv1.1.py DEGLACIAL /home/ATAT/dated_recon.nc MSK
/home/ATAT/experiment1.nc 2 BORDER WEIGHTED INERROR". ATAT then outputs the two maps and a csv
table containing all derived statistics.
Input geochronological data can be created in a GIS environment such as ArcMap or QGIS. Here, the user must
discern the appropriate geochronological data for each grid cell. Since geochronological data is usually stored as
point data, this must be gridded to single grid points as positive values, with surrounding areas of no data assigned
a value of 0. When comparing to a reconstruction (e.g. Hughes et al., 2016), cells outside the reconstruction
should be assigned a value of 0. Those within the reconstruction should be assigned a value corresponding to the
reconstructed age of retreat. The gridded data must be converted to NetCDF format, the details of which are shown
in Table 3. We emphasise that the quality of geochronological data used must be considered, and an example of
how to filter geochronological data is documented in Small et al. (2017). Ice thickness grids can be created using
ice sheet modelling software such as PISM (Winkelmann et al., 2011). The two grids (data and model) must be
aligned and have the same size dimensions for use in ATAT. Examples are included as supplementary material,
including a model output from Ely et al. (in review).
*Acknowledgements:* This work was supported by the Natural Environment Research Council consortium grant;
BRITICE-CHRONO NE/J009768/1. Development of PISM is supported by NASA grant NNX17AG65G and
NSF grants PLR-1603799 and PLR-1644277.We thank Evan Gowan and Lev Tarasov for their constructive
reviews which improved the manuscript.

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

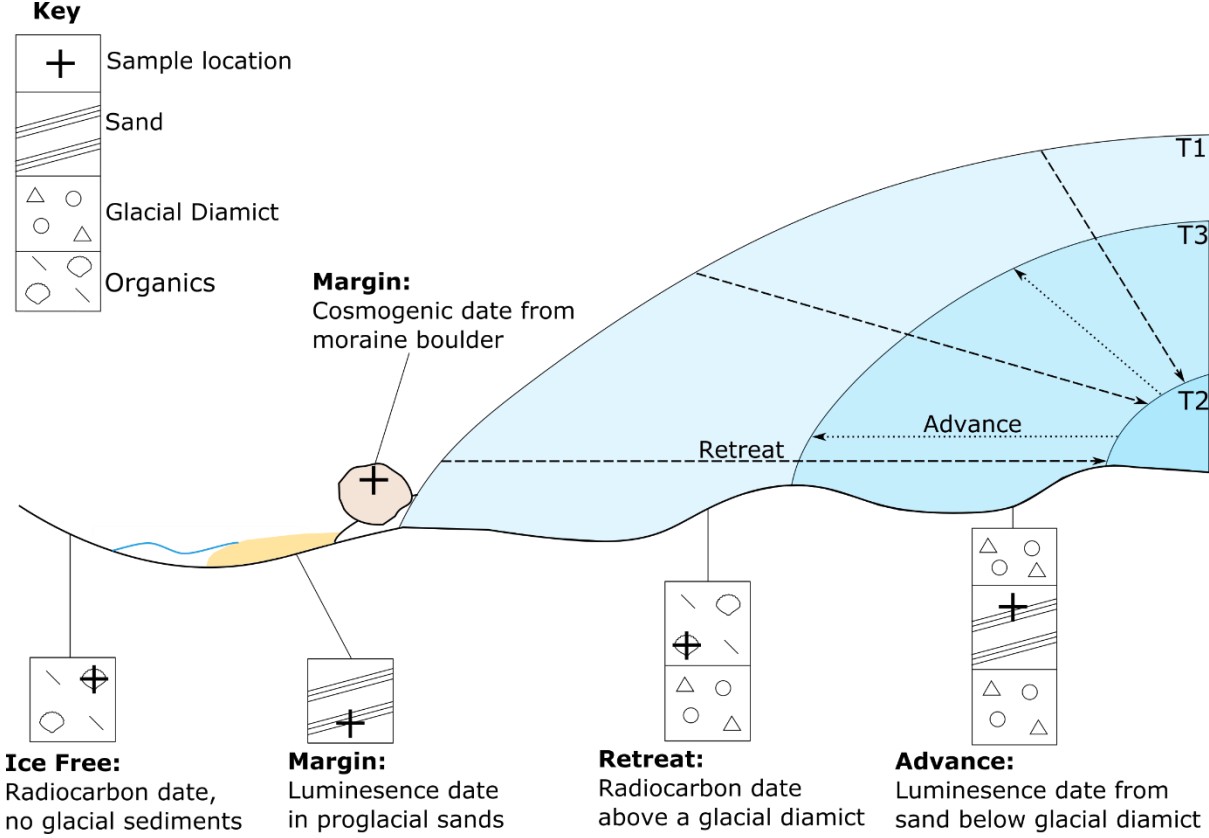

Key

+   Sample location

Sand

Glacial Diamict

Organics

**Margin:**
Cosmogenic date from
moraine boulder

T1

T3

T2

Advance

Retreat

**Ice Free:**
Radiocarbon date,
no glacial sediments

**Margin:**
Luminesence date
in proglacial sands

**Retreat:**
Radiocarbon date
above a glacial diamict

**Advance:**
Luminesence date from
sand below glacial diamict


**Figure 1: Schematic illustration of stratigraphic and inferred glaciological context of geochronological data. Note that**
**at T1 the ice sheet is at its most advanced. It then retreats to a minimum at T2, before re-advancing to T3.**

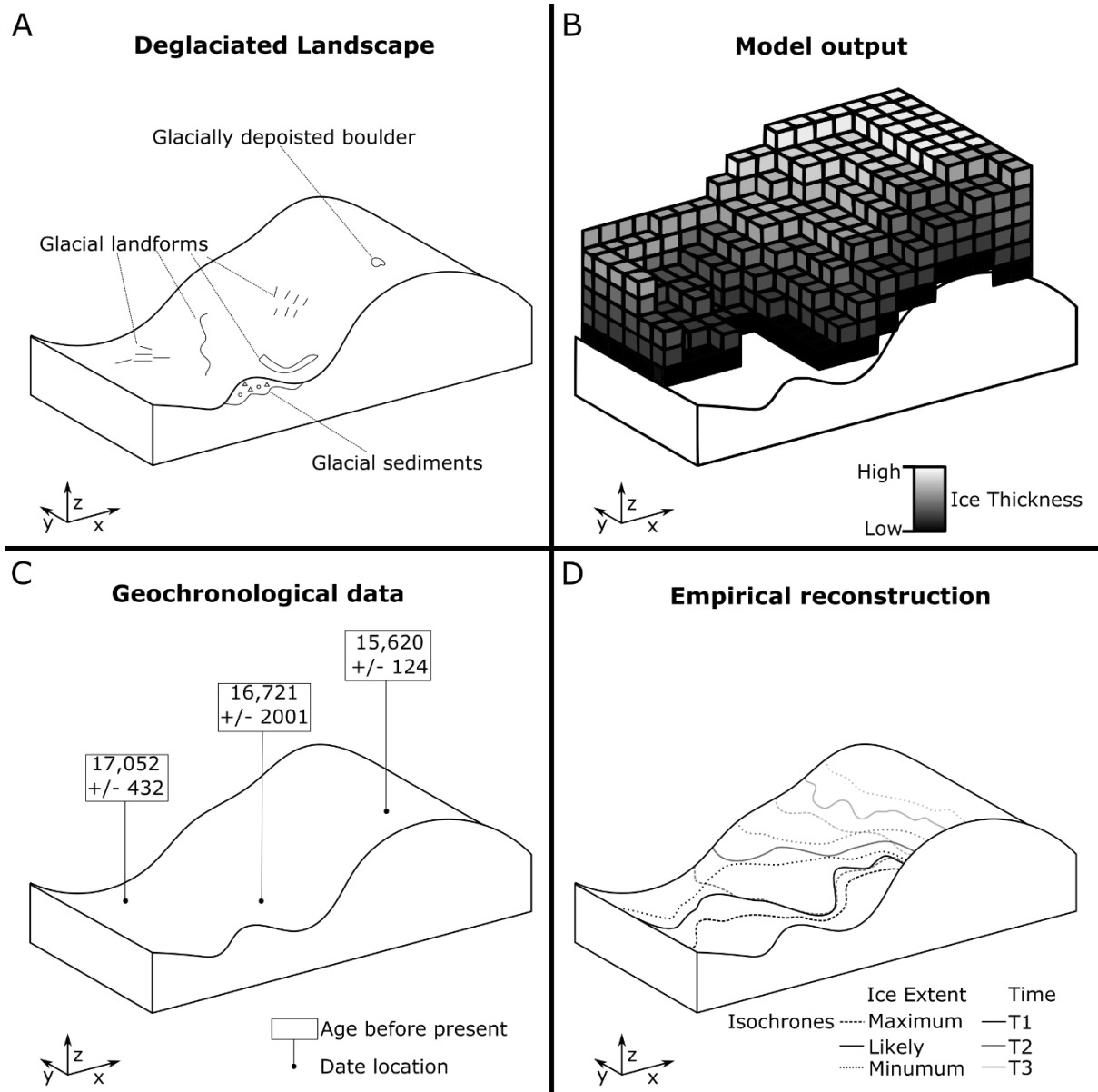

**Figure 2. Schematic of geochronological data and ice-sheet model output. A) A deglaciated landscape,**
**demonstrating some of the features used by palaeo-glaciologists when empirically reconstructing an ice**
**sheet. B) Ice-sheet model output, displaying modelled ice-sheet thickness, in this case at a specific time. C)**
**Geochronological data. D) Empirical reconstruction. Note how the nature of these data vary between**
**source.**

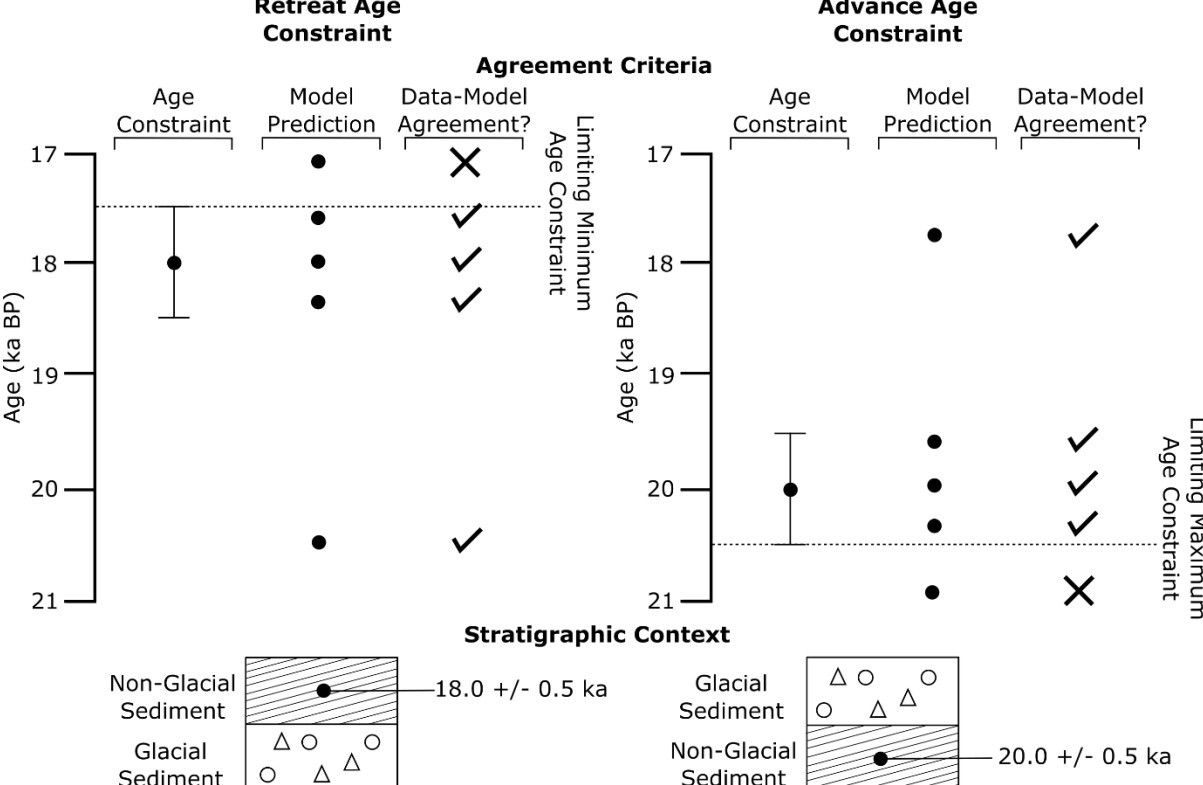


**Figure 3. Schematic of the identification of data-model agreement with consideration of error by ATAT for**
**retreat (left) and advance (right) data. If a model predicts ice free conditions before an ice-free age, or**
**during the associated error, there is data-model agreement. If deglaciation occurs at this location after the**
**error, the model disagrees with the data. If a model predicts ice advance and cover before the advance age**
**and its associated error, there is model-data disagreement. Agreement between the model and data occurs**
**if ice advances over the location after the date, or before the date within the range of the error. This is used**
**by ATAT to categorise sites as to whether agreement or disagreement between the model and data occurs.**

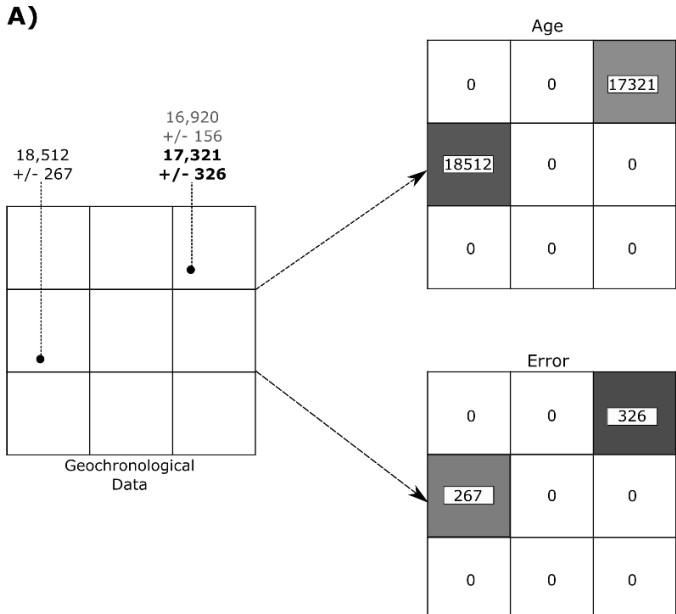

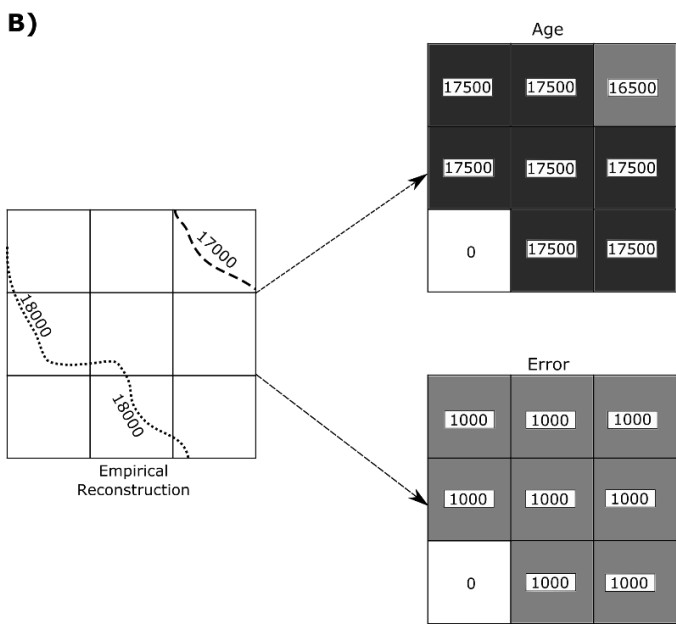


**Figure 4. Examples of empirical data preparation for ATAT. (A) Conversion of geochronological data into a grid for ATAT. In this example the user has made a judgement based on a priori knowledge that the date of 17,321 ± 326 is most representative of the event of interest. Note that age and error are split into separate grids, and that no data regions are assigned a value of 0. (B) Conversion of an empirical reconstruction (margin isochrones) into a grid for ATAT. Here we simply assume that the area between isochrones became deglaciated between at the age between the two isochrones, and that associated error is 1000 years. More complex reconstructions (e.g. Hughes et al., 2016) may require different user-defined rules.**

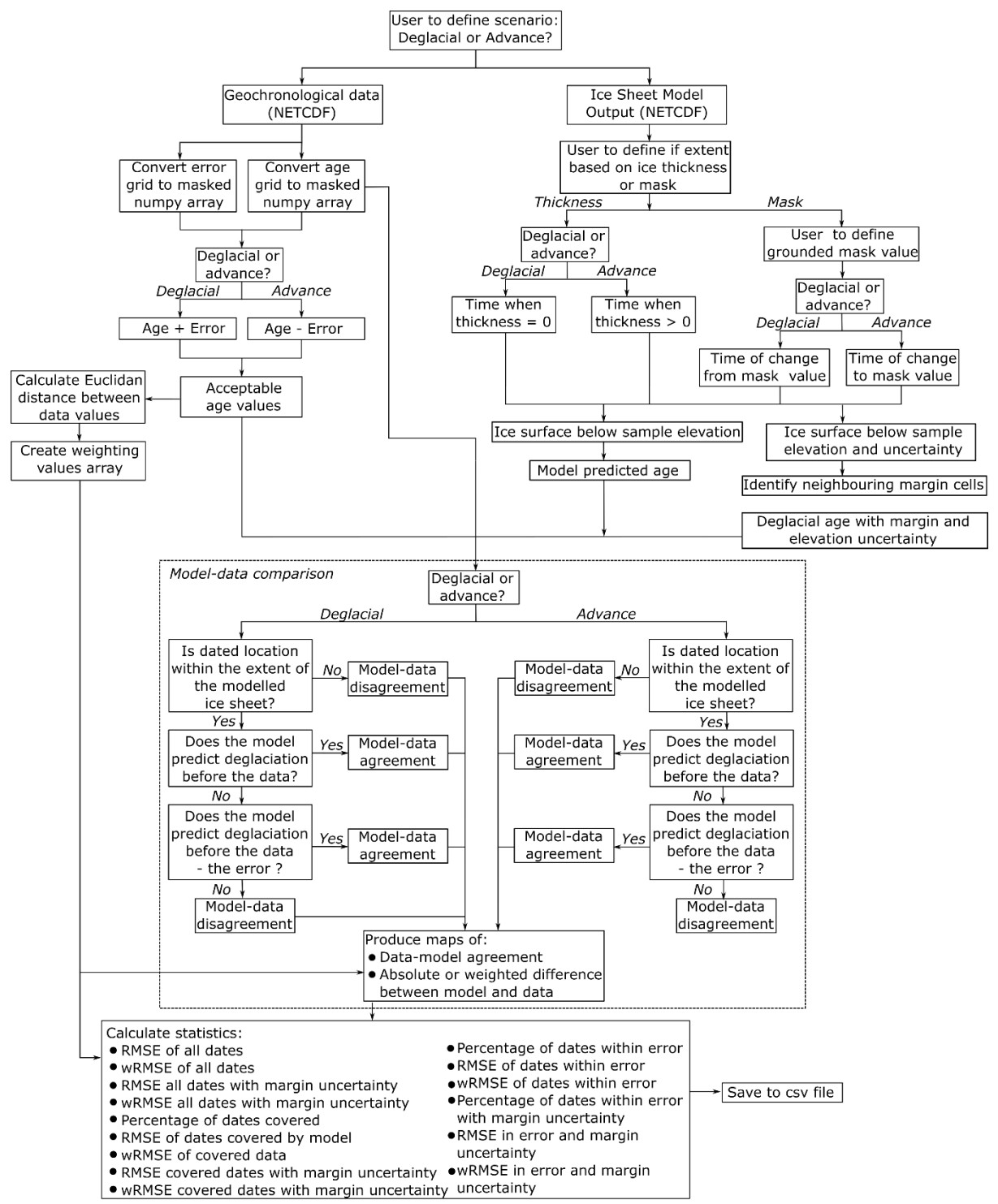

875

876 **Figure 5. Flow chart of ATAT procedure. See text for further description.**

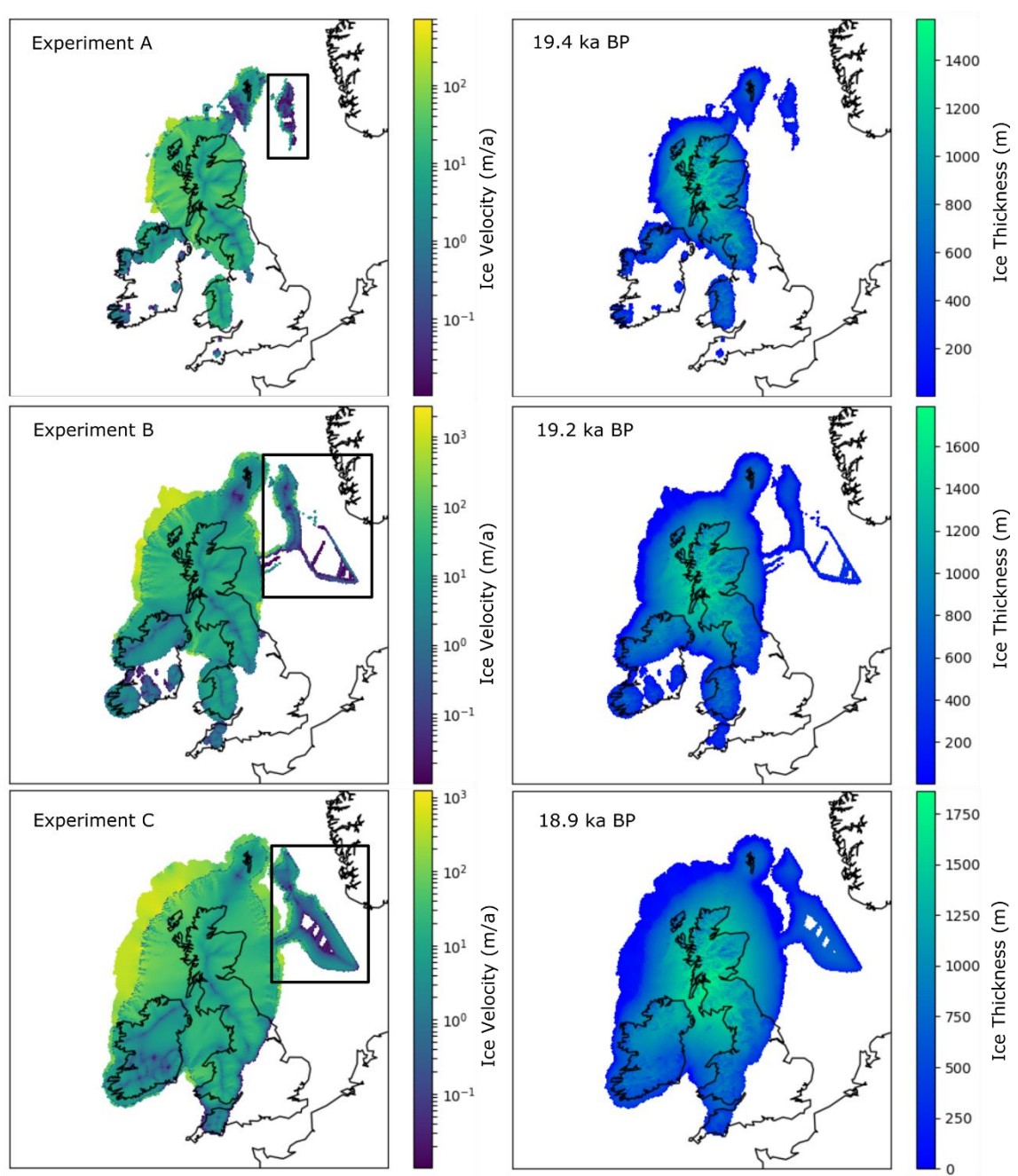

877

**Figure 6. Maximum extent of produced ice sheet for the three experiments. Experiment B is 1°C colder than A, and experiment C is 2°C colder than A. Left panel shows ice velocity, right is ice thickness. The box on the left panel highlights likely erroneous output in the North Sea, likely a consequence of model domain, discussed further in the text.**

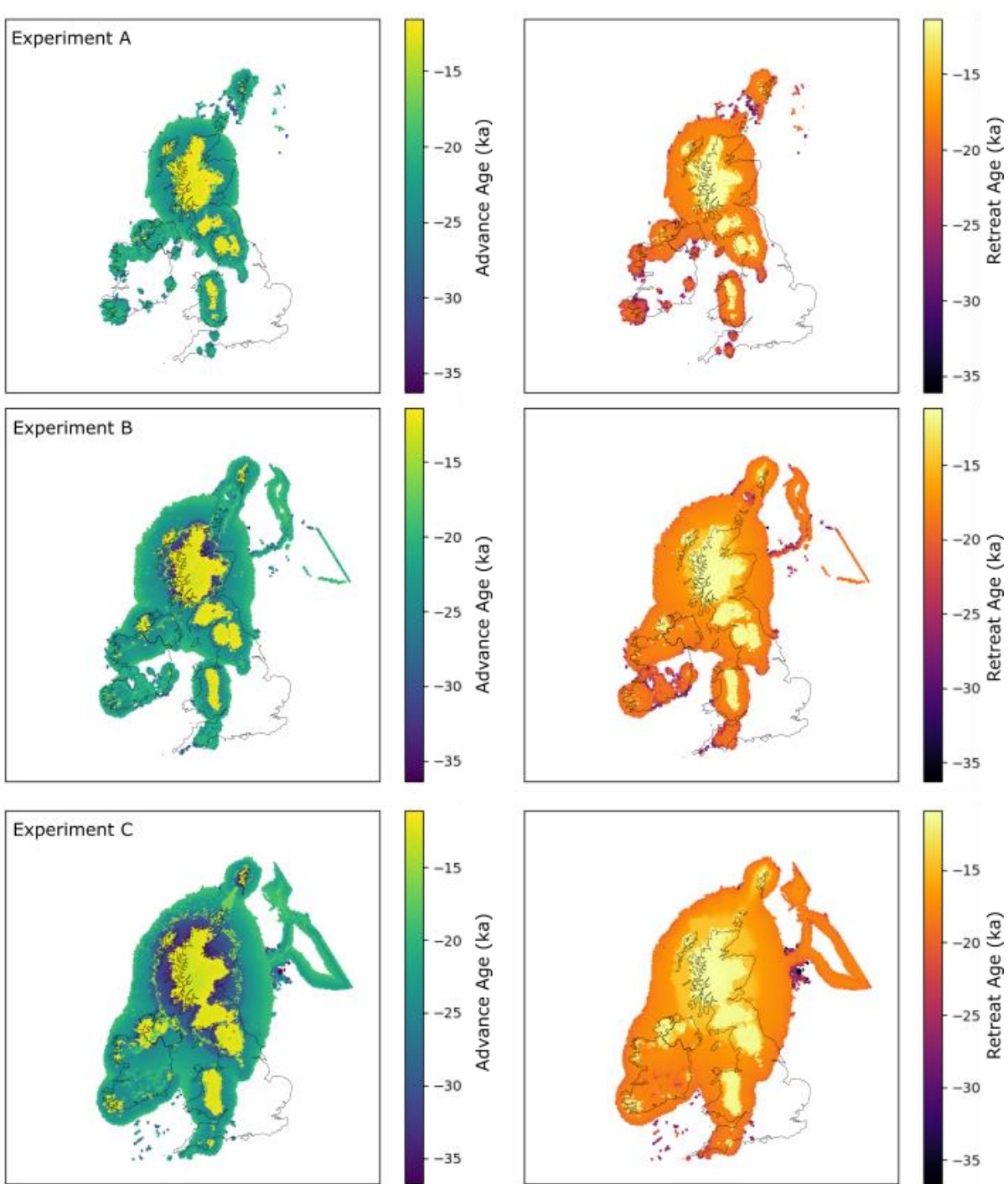

882

**Figure 7. Timing of advance (left) and retreat (right) from the three ice sheet modelling experiments.**
**Experiments are the same as in Figure 6. The early ages toward the centre of the model, and centred over**
**higher topography, represent the modelled extent of the Younger Dryas readvance.**

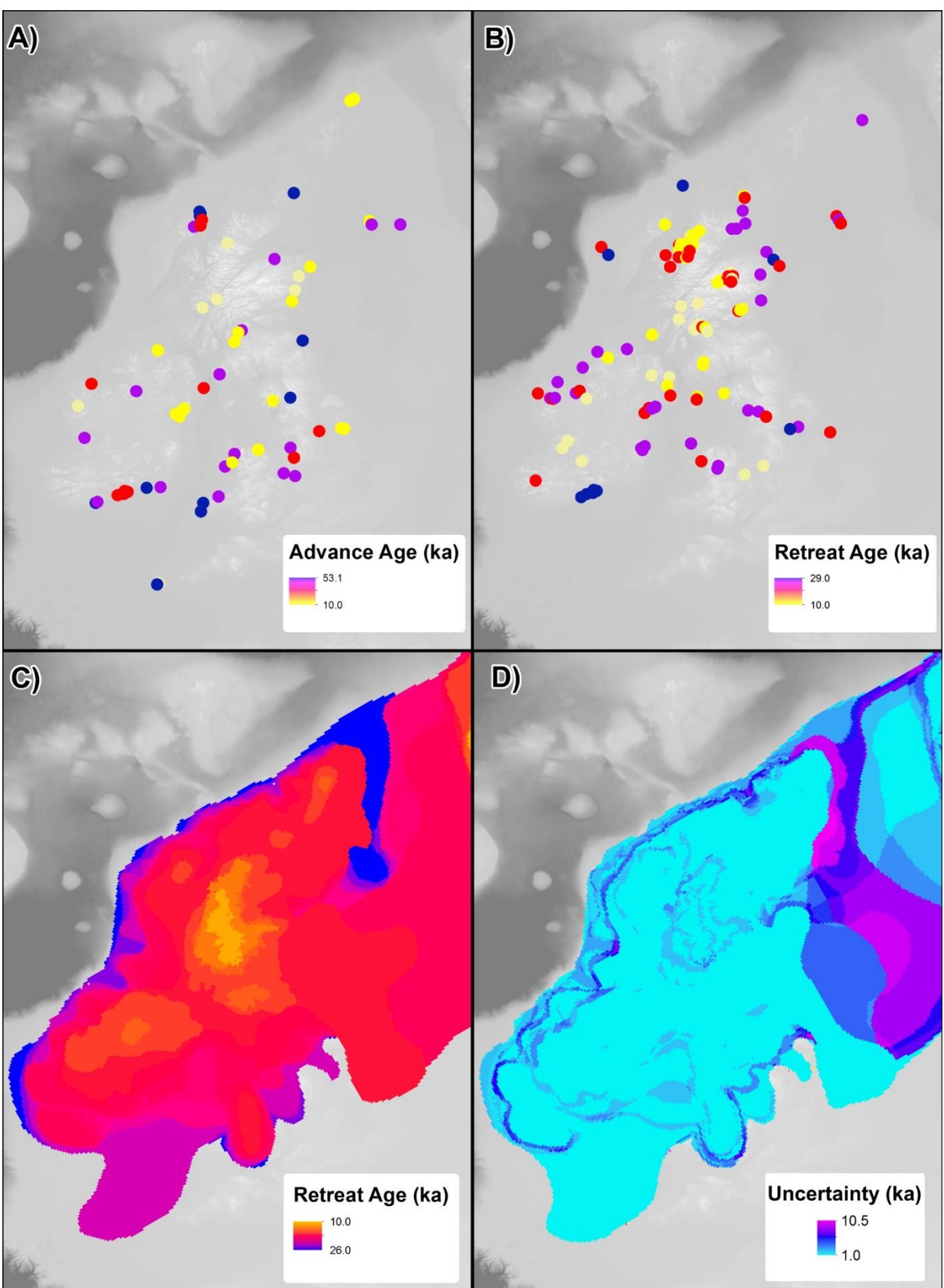

**Figure 8. Example of geochronological data projected onto model raster grids; as point-data in A and B and from an empirical reconstruction in C and D. (A). Advance ages from Hughes et al. (2016). (B) Retreat ages from Small et al. (2017). (C) Retreat age derived from DATED isochrone reconstruction (Hughes et al., 2016). (D) Error associated with reconstruction in C.**

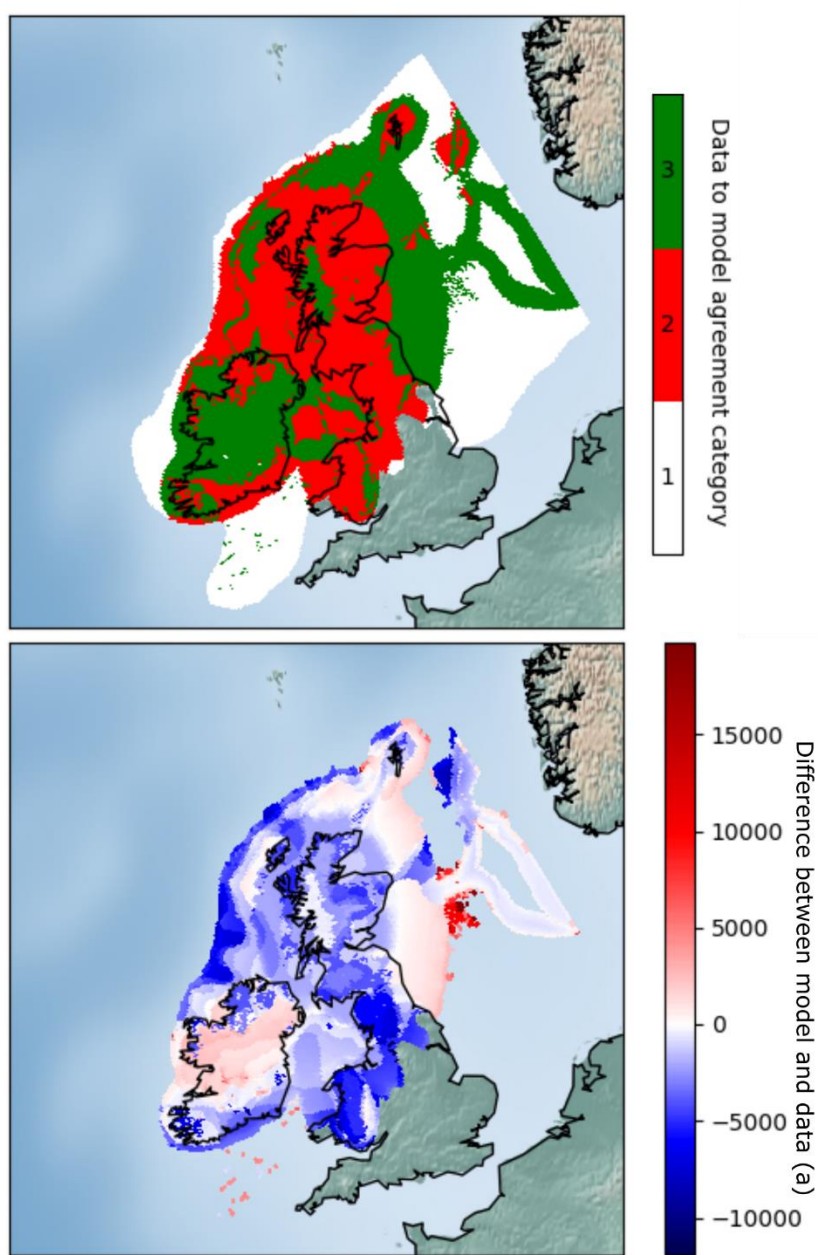

891

**Figure 9. Example mapped outputs from ATAT. In this case, experiment C was compared with the DATED reconstruction. Top map (cumulative agreement) shows categories of data-model agreement across the domain, where 1 = not covered by model, 2 = no agreement and 3 = data-model agreement within error. The lower map (model-data offset) shows magnitude of difference between model and data; negative values show a modelled retreat of ice later than the DATED isochrones, and positive values show a modelled retreat of ice before the DATED isochrones.**

898 Table 1. Classification of geochronological data (after Hughes et al., 2011) and its use in ATAT.

| Class | Glaciological context | Stratigraphic context | Example | Use in ATAT |
|---|---|---|---|---|
| Advance | Ice-sheet build up | Material directly below or incorporated within glacial diamict | Luminescence date from a sand below a glacial diamict | Ice cover a short time after this date |
| Retreat | Ice-free after ice cover | Dated material above glacial diamict | Radiocarbon date of a shell above a glacial diamict | Ice-free conditions from this date onwards (note deglaciation could have occurred a long time before) |
| Ice Free | Ice-free, but lacking direct information regarding ice | Dated material which indicates ice-free conditions but has no relation to ice cover. It may be much younger and not provide much useful constraint. | Radiocarbon date of organic sediments without underlying glacial sediments | |
| Margin | Proximal to an ice sheet margin | Dated material with information that ties it to an ice margin | Luminescence date in proglacial sands | |
| Exposure time (cumulative) | Length of time since sample exposed | N/A | Cosmogenic isotope on erratic boulder above a trimline | Not used |

899

Table 2. Comparison of attributes between geochronological data and ice sheet model output.

| | Nature of data produced | Spatial resolution | Spatial continuity | Temporal frequency and resolution | Sources of uncertainty | Main limitation |
|---|---|---|---|---|---|---|
| **Geochronological data** | Timing of the absence of ice at a location | Point location | Point location, unevenly distributed in space, but can be interpolated | Determined by data availability and associated error | Instrumental, environmental and stratigraphic factors | Reliant upon correct stratigraphic interpretation to tie to glaciological events |
| **Ice-sheet model output** | Simulation of physically plausible ice sheet conditions | Various, ranging from tens to unit kilometres. | Spatially even, regularly-spaced across entire domain | Continuous in time. Precise subannual resolution possible, but not recorded in practice | Parameterisations, boundary conditions | Based upon mathematical and physical approximations of ice flow |

900

| Data source | NetCDF Variable | Units | Dimensions | Description | Notes |
|---|---|---|---|---|---|
| **Ice sheet model output** | Time | Time unit before reference calendar date | x, y | Calendar years before present | |
| | thk | m | time, x,y | Ice thickness | Either "thk" or "msk" required by ATAT. |
| | msk | Integers | time, x,y | Grounded/floating/icefree mask | Either "thk" or "msk" required by ATAT. User defines value referring to the location of grounded ice |
| **Both** | lat | Decimal degrees | x, y | Latitude | |
| | lon | Decimal degrees | x, y | Longitude | |
| **Geochronological data** | age | Time unit before reference calendar date | x, y | Timing of deglaciated conditions | Deglacial and advance ages must be in separate files. |
| | error | Seconds | x, y | Error associated with deglaciated conditions | Error associated with either deglacial and advance age must be in associated separate file. |
| | topg | m | x,y | Modern elevation at resolution of ice-sheet model | |

| elevation | m | x,y | Elevation of collected sample |
|---|---|---|---|

Table 3. Required input variables for ATAT NetCDF files.

Table 4: Example statistics from ATAT. Note that the RMSE is often altered by applying the spatial weighting to create wRMSE.

| | Advance | | | Retreat | | | Empirical Reconstruction; DATED | | |
|---|---|---|---|---|---|---|---|---|---|
| | A | B | C | A | B | C | A | B | C |
| Ice Sheet Modelling Experiment | | | | | | | | | |
| Percentage of dates covered | 52.5 | 72.1 | 88.5 | 76.1 | 91.7 | 96.3 | 32.9 | 52.6 | 69.8 |
| Percentage that agree within error | 65.6 | 72.7 | 72.2 | 22.0 | 22.0 | 12.8 | 23.2 | 27.0 | 17.8 |
| RMSE dates covered by model | 11075.9 | 12732.7 | 13490.3 | 3879.0 | 4180.9 | 4945.4 | 2972.5 | 2678.0 | 2920.8 |
| wRMSE dates covered by model | 13357.3 | 13994.7 | 14849.7 | 4073.4 | 4450.3 | 5165.8 | N/A | N/A | N/A |
| RMSE dates within error | 655.7 | 478.6 | 289.3 | 403.6 | 259.7 | 236.2 | 12023.4 | 10638.7 | 8777.6 |
| wRMSE dates within error | 615.4 | 395.0 | 223.6 | 422.1 | 276.9 | 248.9 | N/A | N/A | N/A |