# Peer review of "ATAT 1.1, an Automated Timing Accordance Tool for comparing ice-sheet model output with geochronological data"

_Geoscientific Model Development, 2018_

## Referee Comment (RC1) · E.J. Gowan (Referee) · 12 Mar 2018

Ely *et al.* present a tool that can be used to evaluate an ice sheet reconstruction or the output of an ice sheet model simulation to chronological data relating to the minimum timing of retreat or maximum timing of advance. I think such a tool is very valuable, and I can see it being useful in my own studies. As stated in the paper, there have been few attempts to directly incorporate individual dates into ice sheet reconstruction evaluation, instead using margin reconstructions such as those by Dyke (2004) and Hughes et al. (2016) for visual comparison. Ely *et al.* use a statistical approach to evaluate whether or not the area covered by the ice sheet reconstruction is consistent

with the chronological information that indicates ice free conditions. As stated in the manuscript, all dates suffer from the "minimum age problem", which is to say that there is an unknown period of time between the retreat of the ice sheet and the age of the material dated (although cosmogenic dates might be close). However, there are few other options to directly evaluate how close the reconstruction is to reality. ATAT is a valuable addition for assessing ice sheet reconstructions that should be used alongside other evaulation methods such as fit to glacial isotatic adjustment indicators.

**1 ATAT software**

Unfortunately, I was unable to get the software to work. I tried to follow the instructions for the format of the NetCDF file as per Table 3, but the program would not accept them, specifically with the geochronological data file. I would suggest adding scripts to build this file, or at least give some example NetCDF files so that it is possible to put things in the right format. It should be noted that the unit "Years before present" is not a valid CF compliant time unit, and will cause command line NetCDF tools like CDO to complain and not work. I would use CF compliant units to make the NetCDF files compatible with other programs.

There is no recommendation on what to do if there are multiple dates in one grid cell. When I was attempting to use the program, I just took the oldest date without regard of the error, but maybe it is better to make a combined probability using a tool like OxCal (Bronk Ramsey, 2009).

Are the errors supposed to be 1-sigma or 2-sigma? The paper does not indicate which should be used. Also, calibrated radiocarbon dates are not normally distributed, what is the recommendation for usage in this program?

Another recommendation I would have is to allow the program to read the required variables (*i.e.* DEGLACIAL/ADVANCE, filenames, THK/MASK) from a file rather than

requiring interactive input. This would greatly streamline usage in scripts where many ice sheet reconstructions are evaluated and plotted automatically.

**2  Paper**

In general, the paper is well written, though I think at times the authors go overboard on detail that is not directly relevant to the tool they are introducing. I think section 2 (background) should be shortened considerably. In the current form, it is almost half of the text. In particular, section 2.2 is a two and a half page review of the inadequacies of ice sheet models. I don't think Geoscience Model Development is really an appropriate venue for such a review, especially since ATAT is not really about fixing these problems. Bringing up these issues here really gives the impression that the authors don't trust ice sheet models at all, which I doubt is the intention. I think anyone doing ice sheet modeling is well aware that it may not be possible to exactly reproduce a configuration that replicates geological observations given the limitations of the models, but they may want to know how close they are!

I think rather than going into such detail on the inadequacies of ice sheet models, it would be more appropriate to detail how ice sheet reconstructions are numerically evaluated at present, such as the extensive Monte Carlo sampling technique used by Lev Tarasov (*e.g.* Tarasov et al., 2012) and evaluations based purely on glacial isostatic adjustment (*e.g* Auriac et al., 2016).

Section 3 does a nice job of explaining the usage of ATAT.

In section 4 and 5, there is a lot of emphasis that this tool be used with large ensemble of model runs. I don't know if this is a realistic outlook if you want to consider realistic climate scenarios. Computing a specific climate state (*e.g* LGM) can take weeks, and a fully coupled ice sheet-climate model is along the lines of months. While ice sheet modelling by itself takes a lot less time to run, I question how valid it is to run a large

ensemble of model runs using a linear scaling of modern day climate. During glacial periods, the ocean and atmospheric patterns were substantially perturbed, and this has follow-on impacts on the growth and retreat of ice sheets. Maybe such an exercise is useful to get a general feel for the kind of climatic conditions are necessary for glaciation, but I don't think it is diagnostic. The discrepancies between the three model runs presented in this section and the chronological data could very well be due to this issue. It could also be related to using a scaling based on the GRIP record, which may not be representative of the climatic variability in the British Iles during the Weichselian Glaciation. None of these points detract from the utility of ATAT, and I think the focus should be more on evaluating the model results. Perhaps one way to do this is to run ATAT using the DATED reconstruction and compare it with one of the model runs. This would illustrate what a good fit looks like.

**3   Minor comments**

- Line 57: I would include Auriac et al. (2016) here.

- Line 301: The sentence here is not complete.

- Line 441: Any reason for using the SPECMAP sea level curve rather than more up to date reconstructions?

- Figure 7: There is no frame of reference in these maps. I'd suggest putting on modern shorelines to make it easier to see what is going on with the model output.

- Figure 8: It is very hard to see the location of the geochronological data on these plots. Maybe it would be better to just plot the raw data as points, rather than plotting them as a grid. I also find it a bit confusing to put both the timing of

advance to the maximum extent and the Younger Dryas readvances on the same plot. I suggest splitting it up into two panes.

Best Regards,
Evan J. Gowan

**References**

Auriac, A., Whitehouse, P.L., Bentley, M.J., Patton, H., Lloyd, J.M., Hubbard, A., 2016. Glacial isostatic adjustment associated with the Barents Sea ice sheet: a modelling inter-comparison. Quaternary Science Reviews 147, 122–135.

Bronk Ramsey, C., 2009. Bayesian analysis of radiocarbon dates. Radiocarbon 51, 337–360.

Dyke, A.S., 2004. An outline of North American deglaciation with emphasis on central and northern Canada, in: Ehlers, J., Gibbard, P.L., Hughes, P.D. (Eds.), Quaternary Glaciations–Extent and Chronology - Part II: North America. Elsevier. Developments in Quaternary Science, pp. 373–424.

Hughes, A.L., Gyllencreutz, R., Lohne, Ø.S., Mangerud, J., Svendsen, J.I., 2016. The last Eurasian ice sheets–a chronological database and time-slice reconstruction, DATED-1. Boreas 45, 1–45.

Tarasov, L., Dyke, A.S., Neal, R.M., Peltier, W., 2012. A data-calibrated distribution of deglacial chronologies for the North American ice complex from glaciological modeling. Earth and Planetary Science Letters 315–316, 30–40.

———————————————

---

## Referee Comment (RC2) · L. Tarasov (Referee) · 23 Mar 2018

After a long description of data and model uncertainties, the authors
present a description and example application of a data-model
comparison software tool. As detailed below, I find several flaws in
the proposed data-model comparison algorithm that need to be
addressed. Furthermore, I do not understand why the authors place so
much attention on model uncertainty in the text and then fully ignore
it in the design of their metrics.  If this tool is meant to be used
by those doing model calibration against paleo observations, then
model uncertainty and downscaling error needs to be explicitly
accounted for in the metric. A few of my issues can be
addressed by making the package more flexible to handle user choices
of metrics (eg implemented by documentation of how to change the
metric).

**major comments**

**I also agree with Evan Gowan's suggestion to significantly shorten**
**the model uncertainty section. When I first read the paper, I**
**expected all the detailed uncertainty discussion to lead to a detailed**
**approach to handling model and data uncertainty. Given that these challenging**
**aspects of model-data comparison are ignored in part (or in whole for model and**
downscaling uncertainty),
**I see no rational for such detailed attention in this paper.**

ATAT requires that geochronological data
346 (advance or deglacial) are interpolated onto the same grid projection and
resolution as the ice-sheet model
347 before use. Though

**Samples for cosmogenic dating in glacial geology contexts are**
**generally gathered in non-singular quantities for a given local (given**
**all the uncertainties with inheritance). Consider a grid cell with**
**two 10Be samples that have very little overlap in their age PDFs. This**
**cannot be represented by a single Gaussian PDF, so I don't see how**
**this interpolated single age approach can work for this context**
**unless non-Gaussian PDF's are permitted (for which there is no indication).**

Preparation of the geochronological data to be the same format and grid resolution
as the ice sheet model output
352 requires use of a GIS software package such as ESRI ArcMap or QGIS. Users must
define deglacial/advance
353 ages based either upon the availability of geochronological data in a cell, or
based upon an empirical
354 reconstruction (Figure 4). Where there are no data (i.e. outside the ice-sheet
limit), the grid value must be kept at
355 0. Given that this may involve many expert decisions (e.g. which date has the
relevant stratigraphic setting,
356 which date(s) are most reliable?), this part of the process is not yet
automated within ATAT.

For a deglacial
367 scenario, a model prediction will be unacceptable if the cell is ice-covered
after the range of the date error is
368 accounted for, but the cell may become deglaciated any time before this.

**what range? one or two or 3 sigma? What if non-Gaussian?**

The opposite is true for advance ages; ice can cover a cell any time after the date

and
371 associated error, but cannot cover the cell before the date of the advance.

**Does this take into account age uncertainty? If so, again to what**
**sigma before rejection?**

To
373 account for the uneven spatial distribution of dates, a weighting for each date
is then calculated based upon their
374 spatial proximity. This weighting is used later when comparing the data to the
model output. To calculate this
375 weighting, the Euclidian distance from each dated cell to its nearest dated
cell (d_i) is calculated. The mean
376 distance between dated cells (dbar) is then calculated, and the weight of each
location (w_i) defined using Eq. (1):

wi=sqrt(di/dbar)

**I don't follow the logic of the weighting scheme. Why should the**
**weight be proportional d_i? What if you have 2 equidistant adjacent**
**cells with dates? Your weighting scheme assigns the same weight to a**
**dated grid cell with one adjacent dated grid cell and a dated grid**
**cell surrounded by equi-distant dated grid cells.**

387 geochronological age and modelled age at each location (Figure 4). Firstly, the
grid cells which have data are
388 categorised as to whether there is model-data agreement, based on the criteria
shown in Figure 3. Since all

**If I follow this correctly, the algorithm is outputting a binary agree-disagree**
result.
**If so, this should be changed to give a continuous metric (that can saturate to a**
large disagree
**result when disagreement is well beyond 3 sigma data + downscaling + some model**
uncertainty.
**Continuous metrics are required for efficient sampling/calibration algorithms.**
You
**will likely start with a bunch of "bad" models, and you need to be able to**
decipher
**which are less bad. If my interpretation of the metric is incorrect, then the**
**description needs to be improved.**

Since all
389 dating techniques only record the absence of ice, geochronological data
provides only a one-way constraint on
390 palaeo-ice sheet activity. For deglacial ages, deglaciation could occur any
time before the geochronological data
391 provided and within the error of the date, but deglaciation must not occur
after the error of the date is considered
392 (Figure 3). For advance ages, advance must have happened after the date or
within error beforehand, but palaeo393
ice sheet advance cannot occur in the time period before that dated error (Figure
3).

Therefore, ATAT also determines the temporal proximity of the geochronological data
and the model
406 prediction. Firstly, a map of the difference between modelled and empirical
ages is created (Figure 5).

```
**Equations 2 and 3 don't take into account dating uncertainty, and**
**are therefore inappropriate.**

**The comparison should also take into account elevation. If the**
**modelled contemporaneous ice surface is below the elevation of the**
**dated sample, then there is datapoint-model consistency even though**
**ice is present contrary to what the presented algorithm would**
**indicate. Given the coarse topography near the present-day margin of**
**Greenland, for instance, elevation needs to be accounted for.**

**The algorithm also lacks consideration of subgrid/downscaling issues. Eg, for an**
**ice marginal gridcell on a 25km grid, one would infer the actual**
**subgrid margin to be somewhere within the gridcell since the next**
**beyond margin gridcell has 0 mean ice thickness, and therefore 0 ice**
**throughout. The easiest way to address this is to have a metric that**
**takes into account proximity of the ice margin as well. This spatial**
**proximity accounting is also important for model calibration to**
**extract a continuous measure that can differentiate between two**
**"bad" models.**
```

different uses. For instance, the percentage of covered dates may prove useful as a
first 421 filter of model runs,
422 whilst the wRMSE of dates within error may be more convenient for choosing
between filtered model runs

```
**So by this logic, a model that was within 1 sigma of all but 2 data points and in**
the rejection
**region for those 2 datapoints (lets say out of 1000 datapoints) would be worse**
than a model
**that was only within 2 sigma everywhere with no data-points in the temporal**
rejection region.
**????**
```

6.1. General Instructions

```
**I would recommend inclusion of an in-line documented sample run**
**script (ie that could be executed with a single command). Model**
**data comparison in generally involve large ensembles, so a script**
**that could be run in a loop would make this more accessible to**
**users.**

**The design of the comparison output needs more thought for use**
**in ensemble comparisons. A summary file should be generated that**
**for each line starts with a model run ID and then includes the**
**summary metric values for that run. The tool should come with**
**an looping script to cycle over model runs from some file list.**

###################
**small corrections**
```

A very large source of uncertainty for modelling palaeo-ice sheets is
the climate used ..  few palaeo-ice sheet models are coupled with 202
climate models
```
**should mention even state-of-the-art GCMs still have relatively**
**large uncertainties for this context (just need to consider the spread across**
PMIP 3 submissions)
```

490 agreement occurs, the RMSE produced are much higher when for the model is compared to the DATED
491 reconstruction.
**English is broken**

Note that a fuller ensemble model of hundreds
509 to thousands of runs is required for full model evaluation (e.g. Hubbard et al., 2009).
**-> of thousands to potentially much more is required...**
**There is no way the uncertainties in a paleo cycle ice sheet model can be**
**honestly represented by even a thousand model runs if one is claiming "full model evaluation"**

---

## Author Comment (AC1) · 10 May 2018

We thank both reviewers for their comments which have helped focus and clarify the manuscript. We have made changes to both the manuscript and the code having considered these comments. Please see attachments for comments.

Please also note the supplement to this comment: https://www.geosci-model-dev-discuss.net/gmd-2018-12/gmd-2018-12-AC1-supplement.zip

---

## Author Response (AR1)

**Authors' response to comments on "ATAT 1.0, an Automated Timing Accordance Tool for comparing ice-sheet model output with geochronological data"**

*Responses in italics.*

*We thank both reviewers for their comments which have helped focus and clarify the manuscript. We have made changes to both the manuscript and the code having considered these comments. On the code, we have incorporated aspects of model uncertainty (margin position, elevation) into the code, and programmed the code in such a way that it can be called from the command-line and is therefore more suitable for batch processing (e.g. of a large ensemble). Changes were substantial enough that we now call this version of the code 1.1.*

*Line numbers below refer to the document which includes track changes.*

**Reviewer 1: Evan Gowan**

General comment

Ely et al. present a tool that can be used to evaluate an ice sheet reconstruction or the output of an ice sheet model simulation to chronological data relating to the minimum timing of retreat or maximum timing of advance. I think such a tool is very valuable, and I can see it being useful in my own studies. As stated in the paper, there have been few attempts to directly incorporate individual dates into ice sheet reconstruction evaluation, instead using margin reconstructions such as those by Dyke (2004) and Hughes et al. (2016) for visual comparison. Ely et al. use a statistical approach to evaluate whether or not the area covered by the ice sheet reconstruction is consistent with the chronological information that indicates ice free conditions. As stated in the manuscript, all dates suffer from the "minimum age problem", which is to say that there is an unknown period of time between the retreat of the ice sheet and the age of the material dated (although cosmogenic dates might be close). However, there are few other options to directly evaluate how close the reconstruction is to reality. ATAT is a valuable addition for assessing ice sheet reconstructions that should be used alongside other evaulation methods such as fit to glacial isotatic adjustment indicators.

*We are glad that the reviewer sees the value in our tool. On their final point, we have now made it explicit in the text that ATAT should be used in conjunction with other evaluation methods, including GIA (lines 47-48 and 553-554).*

1. ATAT software

Unfortunately, I was unable to get the software to work. I tried to follow the instructions for the format of the NetCDF file as per Table 3, but the program would not accept them, specifically with the geochronological data file. I would suggest adding scripts to build this file, or at least give some example NetCDF files so that it is possible to put things in the right format.

*We now have an example NetCDF file in the supplementary material for guidance on how to build the geochronological file.*

It should be noted that the unit "Years before present" is not a valid CF compliant time unit, and will cause command line NetCDF tools like CDO to complain and not work. I would use CF compliant units to make the NetCDF files compatible with other programs.

*The provided example netcdf now uses valid CF compliant units, with a calendar the same as that of the ice-sheet model (in our case years before 1-1-1). We have addressed the calendar issue in the text (Lines 366-367).*

There is no recommendation on what to do if there are multiple dates in one grid cell. When I was attempting to use the program, I just took the oldest date without regard of the error, but maybe it is better to make a combined probability using a tool like OxCal (Bronk Ramsey, 2009).

*The selection of dates for each cell should be left to expert judgement. The issue of data quality is paramount when choosing a geochronological constraint and requires expert judgement – as explored in Small et al. (2017). We have made it more explicit that such expert judgement is needed for individual cells (lines 369-372), and that future attempts should incorporate Bayesian modelling (as per Chiverrell et al., 2013, lines 379-382). In reality, with a high-resolution ice-sheet model (say 5 km) it is unlikely that two equally reliable dates will be contained within a cell – radiocarbon in a core for example should just use the date that is oldest, closest to the glacial contact.*

Are the errors supposed to be 1-sigma or 2-sigma? The paper does not indicate which should be used.

*This is up to the user, and will vary for experimental design. 1 or 2 sigma could also depend upon the source of data for a cell – radiocarbon is typically reported as 2 sigma, OSL as 1. We have reported this in the text (372-375).*

Also, calibrated radiocarbon dates are not normally distributed, what is the recommendation for usage in this program?

*We are using minimum (maximum) constraints for deglaciation (advance), we only look at one side of the distribution (one-tailed constraints as in our Fig. 3). We therefore have an agree/disagree metric that is not dependent upon distribution shape, but rather a user defined acceptable level of error. This is now mentioned in the text (373-375).*

Another recommendation I would have is to allow the program to read the required variables (i.e. DEGLACIAL/ADVANCE, filenames, THK/MASK) from a file rather than requiring interactive input. This would greatly streamline usage in scripts where many ice sheet reconstructions are evaluated and plotted automatically.

*We now enable users to specify all options at the command line, rather than interactively. Scripts could then be developed to batch process several files. The text has been changed throughout to accommodate this change.*

**2. Paper**

In general, the paper is well written, though I think at times the authors go overboard on detail that is not directly relevant to the tool they are introducing. I think section 2 (background) should be shortened considerably. In the current form, it is almost half of the text. In particular, section2.2 is a two and a half page review of the inadequacies of ice sheet models. I don't think Geoscience Model Development is really an appropriate venue for such a review, especially since ATAT is not really about fixing these problems. Bringing up these issues here really gives the impression that the authors don't trust ice sheet models at all, which I doubt is the intention. I think anyone doing ice sheet modeling is well aware that it may not be possible to exactly reproduce a configuration that replicates geological observations given the limitations of the models, but they may want to know how close they are!

*Though we have reduced the length of this section, we think it is important to review these inadequacies of ice sheet models here. We note that whenever ice-sheet models are demonstrated to non-modellers interested in palaeo-ice sheets, they often question why a specific site or geologically recorded event is not accurately replicated. Though people who conduct ice-sheet modelling are aware of the limitations of models, those in the palaeo-community who are do not conduct ice sheet model experiments (half the audience for this paper) are often unaware of model limitations. We note that we also have a lengthy review of the inadequacies of dating, useful for modellers who may not be so close to this discipline. We also disagree that ATAT, and tools like this, won't fix model these problems. Albeit in an indirect way, such comparisons can help. This rationale was stated in the manuscript (299-307), and has been reiterated in the introduction (lines 79-84).*

I think rather than going into such detail on the inadequacies of ice sheet models, it would be more appropriate to detail how ice sheet reconstructions are numerically evaluated at present, such as the extensive Monte Carlo sampling technique used by Lev Tarasov (e.g.Tarasov et al., 2012) and evaluations based purely on glacial isostatic adjustment (e.g Auriac et al., 2016).

*We mention the use of GIA modelling in the introduction and have added the Auriac reference (line 59). Tarasov et al. 2012 run ice sheet models that are not independent of the dated chronology (there is a margin raster, their Fig 2, which "nudges" the ice sheet into place based upon Dykes reconstruction). This calibration is different to model evaluation. We have reduced the uncertainty section, but note that without pointing out the inadequacies of ice-sheet models, we think it would be difficult to make a valid comparison.*

Section 3 does a nice job of explaining the usage of ATAT.

In section 4 and 5, there is a lot of emphasis that this tool be used with large ensemble of model runs. I don't know if this is a realistic outlook if you want to consider realistic climate scenarios. Computing a specific climate state (e.g LGM) can take weeks, and a fully coupled ice sheet-climate model is along the lines of months. While ice sheet modelling by itself takes a lot less time to run, I question how valid it is to run a large ensemble of model runs using a linear scaling of modern day climate. During glacial periods, the ocean and atmospheric patterns were substantially perturbed, and this has follow-on impacts on the growth and retreat of ice sheets. Maybe such an exercise is useful to get a general feel for the kind of climatic conditions are necessary for glaciation, but I don't think it is diagnostic. The discrepancies between the three model runs presented in this section and the chronological data could very well be due to this issue. It could also be related to using a scaling based on the GRIP record, which may not be representative of the climatic variability in the British Iles during the Weichselian Glaciation. None of these points detract from the utility of ATAT, and I think the focus should be more on evaluating the model results. Perhaps one way to do this is to run ATAT using the DATED reconstruction and compare it with one of the model runs. This would illustrate what a good fit looks like.

*We agree that perturbing modern climate by a distal climate record will not capture all of the necessary climate changes. However, this is still done by some palaeo-ice sheet modellers (e.g. Patton et al. 2016 and 2017, Seguinot et al. 2016) to reconstruct these ice masses. ATAT could be used to decipher how well these models simulate the glacial history of an area.*

*However, coupled earth-system models are also being developed which will capture oceanic and atmosphere changes. It will be important to evaluate how close to the data these runs achieve, and where improvement is needed.*

*We have now made it explicit that these 3 model runs are for demonstration purposes only, and our intention is to highlight the utility of ATAT, not to accurately capture climatic conditions over Britain and Ireland through the last deglacial (lines 492-494).*

**3. Minor comments**

Line 57: I would include Auriac et al. (2016) here.

*Auriac et al (2016) now included.*

Line 301: The sentence here is not complete.

*Now fixed.*

Line 441: Any reason for using the SPECMAP sea level curve rather than more up to date reconstructions?

*No. We just had SPECMAP available. As noted, the aim of these simulations are just to provide a bank of 3 simple experiments to compare to geochronological data.*

Figure 7: There is no frame of reference in these maps. I'd suggest putting on modern shorelines to make it easier to see what is going on with the model output.

*Coastlines have been added to this figure.*

Figure 8: It is very hard to see the location of the geochronological data on these plots. Maybe it would be better to just plot the raw data as points, rather than plotting them as a grid. I also find it a bit confusing to put both the timing of advance to the maximum extent and the Younger Dryas readvances on the same plot. I suggest splitting it up into two panes.

*We have plotted the data as points rather than cells for visual clarity. The younger dryas is included to highlight that ATAT only includes the last advance of ice (model could be stopped before younger dryas for a different experiment).*

**Reviewer 2: Lev Tarasov**

After a long description of data and model uncertainties, the authors present a description and example application of a data-model comparison software tool. As detailed below, I find several flaws in the proposed data-model comparison algorithm that need to be addressed. Furthermore, I do not understand why the authors place so much attention on model uncertainty in the text and then fully ignore it in the design of their metrics. If this tool is meant to be used by those doing model calibration against paleo observations, then model uncertainty and downscaling error needs to be explicitly accounted for in the metric. A few of my issues can be addressed by making the package more flexible to handle user choices of metrics (eg implemented by documentation of how to change the metric).

*We hope to have addressed the flaws in the algorithm, many of which we believe to be miscommunication on our part in the paper and addressed by some rewriting of the model-code. These are outlined below. As stated above, we outline model uncertainty to clarify for non-ice sheet modellers (i.e. those who collect geochronological data). We also think it is important to outline this uncertainty when making a comparison tool.*

*ATAT now runs from the command line, to better accommodate batch processing. ATAT outputs all metrics, as they are quick to calculate. This output is shown in the updated Figure 5, which now documents the new metrics designed to account for margin position and vertical uncertainty.*

I also agree with Evan Gowan's suggestion to significantly shorten the model uncertainty section. When I first read the paper, I expected all the detailed uncertainty discussion to lead to a detailed approach to handling model and data uncertainty. Given that these challenging aspects of model-data comparison are ignored in part (or in whole for model and downscaling uncertainty), I see no rational for such detailed attention in this paper.

*We have shortened the length of this discussion, but retain the section as we think that understanding the uncertainty of the model is important when comparing to data. Uncertainty handling will come with ensemble design, the tool asks which ensemble member fits the data best.*

*We have also changed the code to deal with some downscaling uncertainty, in margin position and ice sheet elevation. Our method for dealing with this is now stated on lines 411-422.*

Lines 346-347: Samples for cosmogenic dating in glacial geology contexts are generally gathered in non-singular quantities for a given local (given all the uncertainties with inheritance). Consider a grid cell with two 10Be samples that have very little overlap in their age PDFs. This cannot be represented by a single Gaussian PDF, so I don't see how this interpolated single age approach can work for this context unless non-Gaussian PDF's are permitted (for which there is no indication).

*We addressed the issue of which date to choose for a cell in our response to Reviewer 1, and have strengthened our point, that not all dates are equal and this requires expert judgement, in lines 368-371.*

*It is true that non-gaussian dates occur. Our metrics are based upon whether the model hits a minimum (maximum) constraint in deglaciation (advance), meaning that all geochronological constraints are essentially one tailed depending upon stratigraphic context (see Figure 3). Therefore, the input error is a threshold beyond which model-data agreement does not occur (this is now clarified on lines 371-375. Therefore, if considering a skewed distribution a larger (or smaller) threshold should be defined by the user.*

*Future adaptations may account for more complex treatments of age probability.*

Lines 367-368: what range? one or two or 3 sigma? What if non-Gaussian?

*Our response to the issue of non-Gaussian distributions is stated above.*

*We now state that it is up to the user to define the level of sigma they wish to test (this may be different for radiocarbon, OSL or TCN ages) (lines 372-373).*

Lines 370-371: Does this take into account age uncertainty? If so, again to what sigma before rejection?

*There may be some miscommunication here, as we use the word error to refer to the uncertainty attached to a date (deliberately done to distinguish from model uncertainty). This is specifically input as a variable into ATAT, and we have clarified how to do this.*

Lines 373 and 376 and Eq 1: I don't follow the logic of the weighting scheme. Why should the weight be proportional $d_i$? What if you have 2 equidistant adjacent cells with dates? Your weighting scheme assigns the same weight to a dated grid cell with one adjacent dated grid cell and a dated grid cell surrounded by equi-distant dated grid cells.

*We have changed the spatial weighting scheme to apply a search window which defines a local density of dated cells rather than a nearest neighbour distance. This is now outlined in the manuscript on lines 400-401.*

Lines 387-388: If I follow this correctly, the algorithm is outputting a binary agree-disagree result. If so, this should be changed to give a continuous metric (that can saturate to a large disagree result when disagreement is well beyond 3 sigma data + downscaling + some model uncertainty. Continuous metrics are required for efficient sampling/calibration algorithms. You will likely start with a bunch of "bad" models, and you need to be able to decipher which are less bad. If my interpretation of the metric is incorrect, then the description needs to be improved.

*ATAT outputs several metrics (these are listed at the bottom of Figure 5, and demonstrated at the bottom of Table 4 which seemed to be missing from our original submission). These include both continuous and non-continuous (agree/disagree) metrics. All metrics are output into a .csv file at the end of the comparison, which is named after the simulation name and whether deglacial or advance dates are being tested. Different users of the tool may want to use different metrics in different combinations. For example, to get rid of extremely poor simulations, it might be worth checking the percentage of sites covered. With better simulations, it may be worth checking the wRMSE of sites within dated error. It is also important keep the agree/disagree metric for the following reason: you may do 100 simulations of a palaeo ice sheet and keep getting the same sites that disagree. This may warrant investigation of the erroneous sites and re-evaluation of the data. This logic is stated in the text lines 299-307 and restated is now restated in the introduction (lines 80-84).*

Lines 389-406: Equations 2 and 3 don't take into account dating uncertainty, and are therefore inappropriate.

*We apply the RMSE to all dates to indicate how close to the observations the model is i.e. to develop a continuous metric. We also produce a metric which limits to only those data which have passed the original agree/disagree criteria. This is now clarified in the manuscript (lines 457-459).*

The comparison should also take into account elevation. If the modelled contemporaneous ice surface is below the elevation of the dated sample, then there is datapoint-model consistency even though ice is present contrary to what the presented algorithm would indicate. Given the coarse topography near the present-day margin of Greenland, for instance, elevation needs to be accounted for.

*We agree, and have now included an elevation consideration in the code and in the text. This helps resolve thinning issues for dates on trimlines or possible nunataks. Thank you for suggestion. This is now documented in the manuscript on lines (414-422).*

The algorithm also lacks consideration of subgrid/downscaling issues. Eg, for an ice marginal gridcell on a 25km grid, one would infer the actual subgrid margin to be somewhere within the gridcell since the next beyond margin gridcell has 0 mean ice thickness, and therefore 0 ice throughout. The easiest way to address this is to have a metric that takes into account proximity of the ice margin as well. This spatial proximity accounting is also important for model calibration to extract a continuous measure that can differentiate between two "bad" models.

*This is a great suggestion and something we had overlooked, thank you. We now include a separate metric that accounts for this uncertainty by applying a perimeter surrounding the originally idenfied margin. This is stated in the manuscript on lines 411-414.*

Lines 421-422: So by this logic, a model that was within 1 sigma of all but 2 data points and in the rejection region for those 2 datapoints (lets say out of 1000 datapoints) would be worse than a model that was only within 2 sigma everywhere with no data-points in the temporal rejection region ????

*Apologies, this is a miscommunication on our part. By "first filter" we meant to identify the worst model runs (e.g. those that do not glaciate over say 50% of the dated sites). We have clarified this in the text (lines 462-466).*

I would recommend inclusion of an in-line documented sample run script (ie that could be executed with a single command). Model data comparison in generally involve large ensembles, so a script that could be run in a loop would make this more accessible to users.

*We have redesigned the script to be run from the command line. An example of how to execute the script is included in the script header and in the instructions contained in section 6.*

The design of the comparison output needs more thought for use in ensemble comparisons. A summary file should be generated that for each line starts with a model run ID and then includes the  summary metric values for that run. The tool should come with an looping script to cycle over model runs from some file list.

*A summary output file is produced every time ATAT is run and we have adapted the script to be run from a command line in order that batch processing can be done (e.g. from a shell script).*

Small corrections

Line 201: should mention even state-of-the-art GCMs still have relatively # large uncertainties for this context (just need to consider the spread across PMIP 3 submissions)

*This is now noted (lines 207-208).*

Line 490: English is broken.

*Now corrected.*

Lines 508-509: There is no way the uncertainties in a paleo cycle ice sheet model can be honestly represented by even a thousand model runs if one is claiming "full model evaluation".

*We have rephrased accordingly (lines 553-554).*

[revised manuscript text omitted]

---

## Author Response (AR2)

**Response to major revisions on "ATAT 1.1, an Automated Timing Accordance Tool for comparing ice-sheet model output with geochronological data"**

Reviewers comments are in italics. Previous responses and extracts from the manuscript are in quotation marks with bold and italic font. Line numbers refer to track-changes document.

**Editors comments to the Authors:**

*Dear Jeremy Ely and co-authors,*

*We have now received two further reviews on your manuscript. To my reading, there is one major issue left after this second round, raised by reviewer #2, regarding structural uncertainty. However, the comment by reviewer #1 about the ease of use and the (partial) lack in documentation should be addressed equally. Ideally, the software should be widely usable by the community, but the comment by reviewer #1 indicates it is not so right now.*

*Please provide a response to all issues raised for the next stage of the process.*

*With best wishes,*
*Didier Roche*

We would like to thank the editor for his handling of the manuscript and the two reviewers for providing reviews. These have led to improvements to the manuscript.

Reviewer #2 is concerned that we do not deal with uncertainty in our manuscript. The tool has always addressed data-based uncertainty, but we have further clarified this in the manuscript (Lines 456-458). To avoid confusion over nomenclature, we have explicitly defined three sources of model-based uncertainty (see response to reviewer and lines 191-192). ATAT and the manuscript provide the means to deal with two of these (downscaling and parametric). However, the major or core issue identified by reviewer #2, structural uncertainties introduced by unquantified/unknown processes, remains a challenge for the scientific discipline to tackle and is far beyond the scope of this paper.

Unlike in physics, in palaeo-ice sheet modelling and other Earth Sciences the relationship between data and theory is not straightforward. This complex relationship is exemplified in continental drift, which was proposed in the 1920s but only confirmed by data in the 1960s. For palaeo-ice sheet modelling, there are comparable uncertainties in both the models and the data, and comparison is required for the science to evolve and move forward. ATAT is a step forward for palaeo-ice sheet modelling; it adopts a set of repeatable procedures to compare data and model outputs and permits users and the communities to draw conclusions on how to progress. The procedures in ATAT and discussed in the manuscript account for model-based and data-based uncertainty. However, structural uncertainties introduced by as yet unknown theoretical advances and/or advances in data cannot be accounted for.

We have addressed the concerns of reviewer #1 by adding additional documentation and an example ice grid.

Thank you again for your consideration of this manuscript.

**Reviewer #1**

*I'd like to thank the authors for addressing the comments by myself and Dr. Tarasov. I think the text is in good shape, I have a couple of minor comments below. However, I still think that the documentation of the program is a bit lacking to easily use it. The authors have now included a sample dates file, but did not give an example ice thickness/mask file that would give a template of how to design the NetCDF file. I tried to get some files together myself to run it, but I gave up after a few hours. As an example of my confusion, Table 3 in the text states the dimensions are [x,y], however the program expects coordinates of [x1,y1]. Maybe the program could have the option to detect which coordinates are in the given NetCDF file?*

Thank you for these two thorough and constructive reviews. We have added some text with the aim of helping the reader develop input grids to the general instructions section of the manuscript (lines 602-612), and provided an example ice thickness grid. We have also made updated the reference data to have the more standard coordinate names x and y.

*Minor comments in the text:*

*Line 52: Maybe explicitly name these two tools.*

    The two tools are now named in the text.

*Figure 8: I still think it is extremely hard to make out anything from the age data points in this figure, it looks like a a bunch of tiny red dots. Although I am aware that you are using a high resolution, and you want to show the data at the resolution of the study area, but I still think it would look a lot better to show the data points at a size that the ages can be discerned.*

    The size of the points on Figure 8 have now been increased.

**Reviewer #2**

*Though the revised submission and softwared is an improvement, there remains a core problem. As detailed below, there is a fundamental contradiction in the paper that needs to be addressed. The authors rigorously defend their inclusion of discussion on model and data uncertainties but then ignore these uncertainties in their metrics. For model calibration in the context inferring past ice sheet evolution, a metric needs to take into account all uncertainties, otherwise the resulting inferences will be invalid. And even for the context of just data/reconstruction comparison, inferential uncertainties (measurement, dating, downscaling). need to be accounted for. The two RMSE metrics (equations 1 and 2) could easily be modified to account for observational and downscaling uncertainties.*

    Our responses to reviewer #2 assume the following three categories of model-based uncertainty: (i) downscaling uncertainty – caused by changes in grid resolution when comparing between models and data; (ii) parametric uncertainty – a consequence of uncertain parameters and boundary conditions input into the model; and (iii) structural uncertainty - that which is introduced into a model by a lack of physical understanding of a system (in this case an ice sheet). We define this here, and in the manuscript (lines 191 onward ) to avoid confusion over nomenclature.

The previous iteration of reviews addressed downscaling uncertainty by altering the ATAT code. Parametric uncertainty is often addressed in ice-sheet modelling by conducting ensemble experiments. ATAT can then be used to rule out simulations which perform particularly badly against the data, choose simulations which perform well and quantify the misfit between a single model run and the data. Our means of accounting for parameter uncertainty is therefore to assume that an ensemble of experiments has been conducted, and that ATAT is used to quantify how each individual performs at replicating the timing of ice-free conditions in the geochronological data. Therefore, the handling of parameter uncertainty is left to the user, who should account for this when designing model experiments. This rationale is stated and developed in lines 82-88, 288-295, 473-489, and 568-572.

Since structural uncertainty encompasses all the processes that are poorly understood or unknown to science, a degree structural uncertainty will always remain (unless in the extreme case a system is fully understood, in which case the scientists involved can move onto a different discipline). It is unclear how one would quantify the effects of processes unknown to science within a model. Therefore, that such structural uncertainties exist should be a caveat of any modelling paper, as well as a justification for a scientific discipline to continue to be curious about how a system operates. In the case of ice sheet modelling, it is also unclear how such structural uncertainties caused by unknown processes could be expressed in years, and therefore be comparable to the geochronological data.

Our pragmatic approach would be to compare multiple model outputs, which combined cover plausible ranges of parameters and known processes, to the data, in the way described in ATAT. This may ultimately reveal model-data misfits which mean that structural uncertainties need to be resolved (i.e. we need to better understand this process to fit this data). For example, one could envisage a modelling experiment which included two models, one which does not include a newly implemented process and one which does. Assuming all other things are left equal, ATAT could be used to quantify how introducing an additional process into the model may influence the degree of fit to the model (this idea is also now included in the manuscript (lines 484-489)).

Inferential (data/reconstruction-based) uncertainties are accounted for by ATAT. As listed at the bottom of Figure 5, numerous metrics are calculated by ATAT. Data and model-based uncertainties are included in these metrics. The paper details how it is determined whether a dated site agrees with a model simulation or not (Figures 3 and 5, lines 423-433). This comparison considers the error associated with a dated site and in the latest iteration, elevation and margin uncertainties in the model.  This categorises dates as to their accordance with the model output in question and produces a "Percentage of dates within error with margin uncertainty metric." We then apply the RMSE metrics to only those sites that agree within dating error, modelled margin and modelled elevation uncertainty. In this way, our metrics consider dating and model uncertainty. This is described in lines 456-467, a section we have extended and amended to address the concerns of reviewer #2. Other metrics are also retained, so that users may utilise less stringent metrics. We expect these to be useful in cases where no agreement between data and a model occurs when all uncertainties are considered. A user may then want to distinguish between model runs which get close to matching data, and those that are then far away from the data. This additional point is now included in the manuscript on lines 465-467.

*The user also needs to be provided with clear simple instructions for modifying the metric to do so to insert their own estimates for structural uncertainty. Without these corrections, I suspect ATAT will foster invalid model/data based inferences and thereby do a disservice to the community.*

As noted in the previous response, ATAT assumes that a model has been run multiple times in an ensemble experiment or sensitivity analysis, as occurs frequently in ice sheet modelling (and as is noted in lines 15, 17, 78, 84, 179, 223, 293, 299, 328, 470, 475, 501, 576). Perturbing inputs over a range of plausible values, such ensembles are designed to account, at least partially, for the parametric uncertainty in input parameters for processes implemented in the model. As demonstrated above, this is stated throughout the manuscript, but perhaps most pertinently at lines 178-180, 293-295 and 328-329.

ATAT could also be used on the amalgamation of an ensemble experiment. For example, if a grid showing the mean deglacial age from an ensemble was produced, ATAT could be used to identify how well this mean replicates the data points. One could also envisage considering the uncertainty associated with this modelled age (e.g. a standard deviation of simulations) when producing a modelled uncertainty grid. This and other potential pragmatic uses are now explicitly stated in lines 473-489. Despite these suggestions, the role of this paper is to describe and provide a tool, not an experimental design.

How to account for structural uncertainty, which is essentially unknown until ice sheets are better understood and more (as yet unknown/unquantified) processes incorporated into models, is unclear. Reduction of this uncertainty remains a challenge for the discipline (line 232).

*# detailed comments : responses and revised text*

**"In reality, with a high-resolution ice-sheet model (say 5 km) it is unlikely that two equally reliable dates will be contained within a cell – radiocarbon in a core for example should just use the date that is oldest, closest to the glacial contact."**

*# The expense of 5 km resolution paleo ice sheet models precludes their current useage for the large ensembles needed for paleomodel calibration. Also, in regions with high-topographic variance (eg most of the Greenland margin), there may easily be two relevantly valid dates within 5 km proximity at different elevations.*

We accept that there may be two dates within a cell, and have added the caveat that since we are testing deglaciation, the date considered to be most representative of final deglaciation of a cell should be considered (lines 367-371).

**"Tarasov et al. 2012 run ice sheet models that are not independent of the dated chronology (there is a margin raster, their Fig 2, which "nudges" the ice sheet into place based upon Dykes reconstruction)."** *# Not clear what you mean by "independent", the margin chronology (with uncertainties) is used for nudging the surface mass-balance within climate*

*forcing uncertainties (nudging isn't unbounded), but the amount of nudging then goes into the cost function for the calibration.*

This is a misunderstanding on our part, our apologies to reviewer #2.

*>If this tool is meant to be used by those doing model calibration against paleo observations, then model uncertainty and downscaling error needs to be explicitly accounted for in the metric.*

**"We have shortened the length of this discussion, but retain the section as we think that understanding the uncertainty of the model is important when comparing to data. Uncertainty handling will come with ensemble design, the tool asks which ensemble member fits the data best."**

*# The authors never respond to my concern above about model uncertainty (ie structural uncertainty). They thereby also contradict their own stated requirement in the text: "In order maximise the 68 benefit to all users, any comparisons between palaeo-ice sheet model output and empirical data should ideally 69 consider the inherent uncertainties of both."*

As detailed above, ATAT does consider both data (measurement, reconstruction) and model uncertainty. We consider data uncertainty in the agree/disagree criteria (lines 456-458). Downscaling uncertainty is considered in the elevation and margin uncertainty calculation, added in the previous iteration of review. Parametric uncertainty should be considered in experimental design, as we frequently note throughout the manuscript. Structural uncertainty will always exist to some extent, but is difficult to quantify as noted in our responses above.

*# Furthermore, if the user of ATAT is going to rely on either the original or current descriptions of dating and modelling uncertainty as their main source of understanding these critical issues, then the paper will be doing a disservice to the community. The main message should be that users need to invest the time to really understand these uncertainties or include a collaborator who does.*
*# The provision of a set of in depth appropriate references on this topic would therefore be of much more use than the current or past version cursory discussions.*

We have reiterated that readers should look elsewhere for further understanding of data and modelling issues, and that this is a necessary overview before describing a tool, not a review paper (lines 96-99). This background section has numerous references to relative issues, made shorter on the request of the reviewers in the previous round of reviews. We have also added that collaboration between relevant parties is key (lines 99-101).

**"Uncertainty handling will come with ensemble design, the tool asks which ensemble member fits the data best."**
*# The first phrase is incorrect and the 2nd means the tool is useless for model calibration in the context of inferring past ice sheet evolution. Read Rougier, 2007 to understand what structural uncertainty means and why it has to be included in the likelihood function. The 2nd phrase what are the comparative fits of each ensemble member to the constraint data given all uncertainties.*

As stated above, we show that one can use ATAT whilst model-based uncertainty by conducting an ensemble (we have know of no other practical means exists of handling parametric uncertainty). Acknowledgement to Rougier (2007) has been made to point the reader to a more detailed discussion of model uncertainty in general (lines 98 and 190). We note though that this paper deals with meterological model-data comparison. Although some parallels can be drawn to ice sheet model-data comparison, the data has completely different qualitative and quantitative characteristics, and therefore does not solve the problem of comparing ice sheet models with data.

We reiterate, this is a paper which describes a tool and ways in which it may be useful. This tool and the statements in the paper consider data-based and model uncertainties.

***"It is also important keep the agree/disagree metric for the following reason: you may do 100 simulations of a palaeo ice sheet and keep getting the same sites that disagree."***

*# I don't follow the logic here. What does agree/disagree mean? Your responses seem to put a lot of emphasis on user choice, but here you are imposing a choice on what level of misfit constitutes disagreement instead of having the user apply their "expert judgement"...*

Lines 425-433 describe what we mean by agree/disagree, whether a model and data point both agree on ice-free timing. There is also a figure on this (Figure 3) and a worked example of how ATAT identifies where model-data agreement/disagreement occurs (Figure 9A). We do not impose a choice on what constitutes misfit, we leave this to the user as they can define the level of data error in the input files. This is described in lines 376-379.

***"Three classes of data are of particular use for constraining palaeo-ice sheets; 46 (i) geomorphological data, (ii) relative sea level history, and (iii) geochronological data"***
*# This list is too limited. Present-day rates of vertical uplift are also a powerful constraint (cf my 2012 paper) for the North American and Eurasian ice sheets. So change RSL -> geophysical constraints (including RSL and present-day vertical velocities).*

This has been amended in the text accordingly (lines 47 and 56-59).

***"applying offsets derived from ice core records to contemporary climate (Hubbard et al., 2009) and scaling 202 between present-day conditions and uncoupled global-circulation-model simulations at maximum glacial 203 conditions (Gregoire et al., 2012; Gasson et al., 2016)."***
*# The usual standard for example references is either a recent detailed review or first use. The above citations do not follow either logic.*

In the absence of a review of palaeo-ice sheet modelling climate forcing techniques, we have added additional reference to early use of the methods.

***"Since all dating 390 techniques only record the absence of ice, geochronological data provides only a one-way constraint on palaeo-391 ice sheet activity. For deglacial ages, deglaciation could occur any time before the geochronological data provided 392 and within the error of the date,"***

*# Incorrect. As detailed:*

We are unsure how this is incorrect, as we are describing how all deglacial (advance) dates are minimum (maximum) constraints (see below).

*Figure 3*

*# I am concerned about the metric indicated in this figure. It seems to indicate that a model with say last retreat for a given grid cell at say 30 ka, will be given the same score for this site as for a model that retreats ice at say 19 ka (for the given sample date of 18 +/- 1 ka). If my interpretation is correct, this needs to be remedied. This should be obvious for eg cosmogenic dates where inheritance will make the date if anything too old (and therefore cosmogenic dates may be maximum limiting depending on type of sample and location). But even for C14 dates, where the issues of sample availability, time required for in-migration of plants,.. mean that the dates are generally minimum limiting, I can't see anyone saying that a model with a 12 kyr misfit with a minimum limiting age should score the same as a model that fits the sample within sample age uncertainties.*

Firstly, this is a schematic for deriving whether an age agrees or disagrees with a model, whilst accounting for error of the age, not a metric in itself. Closeness of fit is assessed later. Clarification of this has been made in the figure caption. Secondly, the statement that the example given would provide the same score is incorrect. The 30ka retreat would be identified as model-data agreement (strictly this is acceptable given that all deglacial ages are treated as minimum constraints) but the RMSE scores would be heavily effected by a difference of 12ka, and this outlier will be evident on the produced model-data offset map (e.g. Figure 9B). We are aware of the issue  and it is already described in the manuscript (lines 309-317) with the suggestion that loose constraints are removed from comparison data by initial data filtering (lines 372-374). Thirdly, cosmogenic dates can be considered as minimum limiting. There are processes that make a cosmogenic date too young, for example vegetation/sediment cover and post glacial depositional erosion. Erosion specifically is more likely to affect all samples from a site to a broadly similar degree and the erosion rate must be assumed in calculating an exposure age. Erosion rates are generally assumed to be low (e.g. 0 -1 mm ka$^{-1}$) but higher rates (for whatever reason) would cause the apparent exposure age to be too young. While inheritance can be an issue, it is common practise for multiple samples to be considered in cosmogenic dating and ages that clearly exhibit inheritance can be excluded (the reference of Small et al. 2017 which we repeatedly refer to details this further). Conversely, unquantified error introduced by an incorrect assumption of erosion rate would potentially affect all ages from a site to a broadly similar degree and thus said ages may not warrant exclusion.

*# equations 1 and 2 are highly problematic given that observational and model uncertainties are ignored. And this again contradicts the authors own statements about the importance of these uncertainties.*

Observational uncertainties are not ignored in the calculation of these metrics, as they are applied to categories of data according to whether model-data agreement occurs when considering the associated error of an age (this is now more explicitly stated in lines 462-467. and on Figure 5). Downscaling (margin and vertical) uncertainties of the model are now accounted for in ATAT (due to previous reviews, stated on lines 410-421). Parametric uncertainties can be overcome to some extent by ensemble experiments, and this is stated in the text (see previous responses and lines 475-478).

[revised manuscript text omitted]
 here as all uncertainty that arises within a model due to a lack of physical understanding of the system in question. In this broad sense, structural uncertainty encompasses all processes which are not incorporated in a model. This may include some processes which are well understood, but not included in a model due to the lack of a numerical formulation, for computational efficiency, or because they are deemed unimportant for the question being studied. In a broader sense, structural uncertainty also includes processes that are as yet unknown to science and therefore are not implemented in a model. Reducing structural uncertainty, by including additional pertinent processes in models, is an ongoing challenge for ice-sheet modelling.

There is another structural uncertainty cause of which hinders ice-sheet models not 
[revised manuscript text omitted]

---

## Author Response (AR3)

Dear Dr Roche,

Thank you for your handling of our manuscript, and constructive comments. Please see response to your points below. We hope to have addressed your concerns, which may be due to poor communication on our part.

*I have read in depth your manuscript, the concerns expressed by reviewer #2 and your response to reviewers.*

*I still find two aspects problematic:*

*1-/ Reviewer #2 highlighted that the RMSE metrics on which you base the ATAT tool needs to include for observational and downscaling uncertainties. Though you discuss at length in your response to reviewers the different sources of uncertainties and their origin, you did not reply to this specific request in positive or negative. I am sure the reviewer is well aware that structural uncertainty is present in any modeling work so this is not the point for discussion. You advocate for the user of your tool to have run ensemble experiments of their ice-sheet model which I understand, but how and where does these ensemble runs come into the uncertainty in your tool? Do you consider running your tool for each of the member independently and accounting for such uncertainty a posteriori? But how? I feel that the request of reviewer #2 in this context regarding equations 1&2 is a valid one, unless I miss part of the reasoning of course.*

ATAT does include a procedure which accounts for observational and downscaling uncertainties, though this is not reflected in equations 1 and 2 alone. ATAT categorises the dated-locations using the criteria of modelled ice cover, agreement within dating (observational) error and those within margin uncertainty (downscaling uncertainty). By applying equations 1 and 2 to these categories of dates, observational and downscaling uncertainties can be accounted for. We have clarified this in the text (lines 469-473). Note that ATAT also outputs the percentage of dates in each category, so that models which do not agree with many sites can be identified (list at end of Figure 5).

We chose to retain these categories, rather than produce a single metric, so that a user may select the appropriate metric for their model experiment.

We advocate applying the tool to members of an ensemble (lines 487-490) so that members may be ranked or a weighting scheme applied when calibrating a model ensemble. An alternative is also suggested in the text, that ATAT be applied to a model with an independently derived uncertainty distribution of deglacial timing at each model cell. For example, an ensemble could be run producing a mean deglacial timing and a standard deviation of deglacial timing. The user could then use the extremes of this modelled uncertainty to test which dates agree within dating error and margin uncertainty (lines 491-494).

Perhaps some of this concern comes from the difference between our approach and that of Reviewer 2. As many of us come from a data-based background, we know that some uncertainties are poorly defined. In our case, all data collected relies upon stratigraphic interpretation. This is why we are keen to retain the discussion of data uncertainty (Section

2.1), and to air the possibilities of also using the model to identify data outliers (lines 495; 580-584; 596-597).

*2-/ In the new paragraphs that have been implemented in your manuscript I read: "Structural uncertainty is related to parametric uncertainty, but has a broader remit, and is defined here as all uncertainty that arises within a model due to a lack of physical understanding of the system in question." pp line and following. Does this means you have, for this particular manuscript, specific definition of structural uncertainty which is not the common one? Could you highlight the reason for doing so and what is the exact difference with the common definition?*

Having read some more literature around structural uncertainty, we have refined the definition stated and added reference to the relevant literature. In geoscience it seems that structural uncertainty is that related to code structure, which processes are included and how. Confusion was introduced as other disciplines have a different definition. We have also noted a means for accounting for this uncertainty (though it is not commonly done in palaeo-ice sheet modelling), by conducting a multi-model comparison including models with different process formulations (lines 225-233).

[revised manuscript text omitted]